# The sponge microbiome within the greater coral reef microbial metacommunity

Daniel F.R. Cleary [1,2,3], Thomas Swierts[4,5], Francisco J.R.C. Coelho[1,2], Ana R.M. Polónia [1,2], Yusheng M. Huang[3,6], Marina R.S. Ferreira [1,2], Sumaitt Putchakarn [7], Luis Carvalheiro [2], Esther van der Ent [4,5], Jinn-Pyng Ueng[3,8], Newton C.M. Gomes [1,2] & Nicole J. de Voogd[4,5]

Much recent marine microbial research has focused on sponges, but very little is known about how the sponge microbiome fits in the greater coral reef microbial metacommunity. Here, we present an extensive survey of the prokaryote communities of a wide range of biotopes from Indo-Pacific coral reef environments. We find a large variation in operational taxonomic unit (OTU) richness, with algae, chitons, stony corals and sea cucumbers housing the most diverse prokaryote communities. These biotopes share a higher percentage and number of OTUs with sediment and are particularly enriched in members of the phylum Planctomycetes. Despite having lower OTU richness, sponges share the greatest percentage (>90%) of OTUs with >100 sequences with the environment (sediment and/or seawater) although there is considerable variation among sponge species. Our results, furthermore, highlight that prokaryote microorganisms are shared among multiple coral reef biotopes, and that, although compositionally distinct, the sponge prokaryote community does not appear to be as sponge-specific as previously thought.

[1] Department of Biology, University of Aveiro, Campus de Santiago, 3810-193 Aveiro, Portugal. [2] CESAM, University of Aveiro, Campus de Santiago, 3810-193 Aveiro, Portugal. [3] Tropical Island Sustainable Development Research Center, National Penghu University of Science and Technology, 300 Liu-Ho Rd., Magong City, Penghu 880, Taiwan. [4] Marine Biodiversity, Naturalis Biodiversity Center, PO Box 9517, 2300 RA Leiden, The Netherlands. [5] Institute of Environmental Sciences (CML), Leiden University, PO Box 9518, 2300 RA Leiden, The Netherlands. [6] Department of Marine Recreation, National Penghu University of Science and Technology, 300 Liu-Ho Rd., Magong City, Penghu 880, Taiwan. [7] Institute of Marine Science, Burapha University, Chon Buri 20131, Thailand. [8] Department of Aquaculture, National Penghu University of Science and Technology, 300 Liu-Ho Rd., Magong City, Penghu 880, Taiwan. These authors contributed equally: Daniel F.R. Cleary, Thomas Swierts. Correspondence and requests for materials should be addressed to D.F.R.C. (email: cleary@ua.pt) or (email: dfrcleary@gmail.com)

I n recent years, high-throughput sequencing methods have generated an unprecedented amount of information on the structural and functional diversity of microbial communities[1]. Marine host-associated prokaryote communities, particularly those associated with sponges, have been reported to be highly diverse[2]. Despite the constant influx of seawater, sponges are able to sustain dense and diverse symbiotic communities, which can comprise up to 35% of sponge biomass[3,4]. These associations, furthermore, appear to be consistent over different geographical areas and under different environmental conditions[5–10].

Much like the human gut, sponges are considered to be an important model to study host–prokaryote associations[4]. Although much recent research has characterised the phylogenetic diversity and biogeography of sponge-associated microorganisms, relatively little is known about a range of other hosts in coral reef ecosystems. If, and to what extent, sponge-associated microorganisms occur in these other hosts is still largely unknown. This is an important hiatus in our understanding of coral reef microbial ecology given that the prokaryote communities of sponges are part of a wider prokaryote 'metacommunity' of host-associated and free-living (in sediment and seawater) microorganisms[11]. This metacommunity forms the regional pool of prokaryote species from which local (within a single host) host-associated communities of microorganisms are assembled. These local communities are presumably linked by dispersal, mainly between host organisms and the external environment, thus maintaining the intricate structure of the metacommunity[12]. Occasionally, direct contact between different host taxa may also induce dispersal and shape the microbial community. Pratte et al.[13], for example, showed that direct contact between turf algae and the coral species *Porites* sp. had a strong influence on the coral (but not the algal) bacterial community.

In the present study, we assess and compare prokaryote communities from a range of host taxa and the abiotic environment (sediment and seawater) in Indo-Pacific coral reef habitats. Our samples include high and lower diversity hosts. High diversity hosts include samples of algae, chitons, stony corals and the sea cucumber gut and mantle. Samples from these hosts are compositionally similar, and have relatively high abundances of operational taxonomic units (OTUs) assigned to the phylum Planctomycetes and relatively high OTU richness and evenness. Prokaryote communities of this group also share significantly more OTUs$_{100}$ (OTUs with >100 sequences) with sediment (i.e. OTUs found in sediment but not seawater) than other biotopes. The lower diversity host group includes sponges, sponge denizens and the nudibranch gut and mantle biotopes. Compared to the first group, samples of this group have a relatively low OTU richness and evenness (with the exception of high microbial abundance [HMA] sponges) and a relatively low percentage of sediment OTUs$_{100}$. The mean percentage of total environmental OTUs$_{100}$ (OTUs recorded in sediment and/or seawater), however, is highest in sponges. The main compositional differences observed in the present study appear to be driven by the apparent permeability of certain taxa (namely algae, sea cucumbers and stony corals) to sediment prokaryotes and the concomitant high prokaryote richness found in these taxa. In turn, sponges, nudibranchs, flatworms and sponge denizens have much fewer sediment prokaryotes OTUs$_{100}$ and a concomitantly lower prokaryote richness, despite having a sometimes very high contribution of environmental OTUs$_{100}$ to total OTUs$_{100}$ richness.

## Results

**Approach**. In this study, we applied high-throughput 16S rRNA gene sequencing analysis to simultaneously assess the diversity of 216 prokaryote communities (Supplementary Data 1) from the following 14 biotopes: algae, chitons, stony corals, sea cucumber gut, sea cucumber mantle, sponge denizens (organisms that live on or within sponges), flatworms, nudibranch gut, nudibranch mantle, soft corals, sponges, sea urchins, seawater and sediment (Fig. 1). All host-associated biotopes consisted of multiple species, with the exception of chitons (only included the species *Liolophura japonica*), soft corals (only included the species *Cladiella* sp.) and sea urchins (only included the species *Diadema savignyi*). Samples were collected from coral reef sites in Taiwan and Thailand (Supplementary Data 1).

**General patterns**. We recorded 30,725 OTUs assigned to 68 phyla over 2,160,000 sequences (after rarefying to 10,000 sequences per sample). The number of OTUs recorded per sample varied from only 103 for a gut sample of the nudibranch *Phyllidia picta* to 3704 for a sediment sample (Supplementary Data 1). The richest host-associated sample (2997 OTUs) was from the gut of the sea cucumber *Holothuria hilla*. The richest (in terms of OTUs) and most abundant (in terms of sequences) prokaryote phyla sampled in the present study included Proteobacteria, Bacteroidetes, Planctomycetes, Acidobacteria, Chloroflexi and Actinobacteria. Abundant phyla with relatively few OTUs, but numerous sequence reads, included Tenericutes, Cyanobacteria, Spirochaetae, Thaumarchaeota and Nitrospirae (Supplementary Fig. 1 and Supplementary Table 1).

The relative abundance of 18 of the most abundant phyla (with the exception of Proteobacteria) and the four most abundant proteobacterial classes (with the exception of Gammaproteobacteria), varied significantly among biotopes (Fig. 2; pairwise comparisons between pairs of biotopes are presented in Supplementary Data 2). Some biotopes were strongly enriched by specific prokaryote phyla. The abundance of Planctomycetes, for example, was significantly higher in sediment, and the sea cucumber gut and mantle than the nudibranch gut and mantle and sponge biotopes (Fig. 2i and Supplementary Data 2). The relative abundance of Chloroflexi, in turn, was highest in the sponge, sponge denizen and nudibranch mantle biotopes and significantly higher than in the algae and nudibranch gut biotopes. There was, however, pronounced variation in Chloroflexi abundance within these biotopes as shown by the large standard deviations in Fig. 2d. For example, the sponge species *Aaptos lobata*, *Hyrtios erectus* and *Xestospongia testudinaria*, which have been previously identified as HMA sponges or have been shown to house prokaryote communities very similar to those found in HMA sponges[7,14–18], had higher relative abundances of Chloroflexi, and other taxa including SBR1093 (Fig. 2p) and Poribacteria (Fig. 2r), than all other sponge species (Supplementary Data 1). At the class level, alphaproteobacterial abundance was highest in the nudibranch mantle and significantly higher than in the sea cucumber gut, soft coral, sponge and sea urchin biotopes (Fig. 2t and Supplementary Data 2). Deltaproteobacterial abundance was highest in the stony coral, sea cucumber gut and mantle, sediment and sea urchin biotopes and significantly higher than in the algal, sponge denizen, nudibranch gut and mantle, flatworm, soft coral, sponge and seawater biotopes (Fig. 2u). Betaproteobacterial abundance was highest in the sponge and sponge denizen biotopes and significantly more so than in the algae, sea cucumber gut and nudibranch gut and mantle biotopes (Fig. 2v and Supplementary Data 2).

OTU sample richness was highest in the sediment, chiton, algae, stony coral and sea cucumber gut and mantle biotopes and lowest in the flatworm, sponge, nudibranch gut and mantle, soft coral, sea urchin and seawater biotopes (Fig. 2x and Supplementary Data 1). This same pattern also applied to cumulative OTU

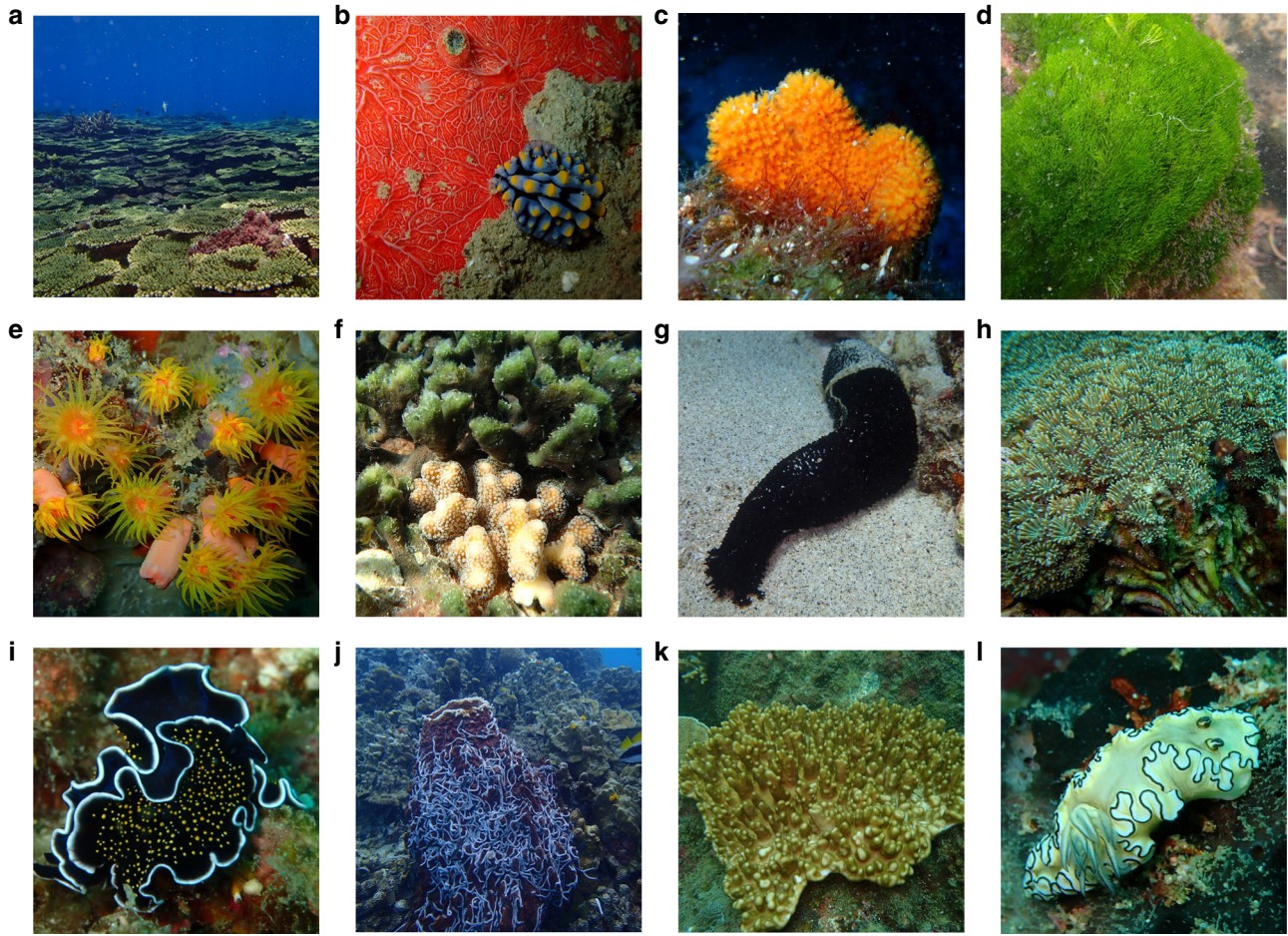

**Fig. 1** Pictures of sampling sites and organisms sampled during the present study. **a** Coral reef in the southern Penghu islands, Taiwan, **b** the nudibranch *Phyllidia* cf. *coelestis*, **c** the sponge *Ptilocaulis spiculifer*, **d** the green alga *Chlorodesmis fastigiata* in shallow water, **e** the sun coral *Tubastraea coccinea*, **f** the green sponge *Haliclona cymaeformis*, **g** the sea cucumber *Holothuria leucospilota*, **h** the stony coral *Galaxea astreata*, **i** the spotted flatworm *Thysanozoon nigropapillosum*, **j** the barrel sponge *Xestospongia testudinaria* covered by sea cucumbers (*Synaptula* sp.), **k** the soft coral *Cladiella* sp. and **l** the nudibranch *Doriprismatica atromarginata*. All photographs were taken by D.F.R. Cleary or N.J. de Voogd

richness (Supplementary Fig. 2). Histograms of OTU richness also showed largely non-overlapping distributions with samples of sponges and the nudibranch mantle clustered at low OTU richness values while samples of algae, the sea cucumber gut and sediment were spread out over a larger range at higher OTU richness values (Supplementary Fig. 3). This distinction also held after removing all OTUs <100 sequences (Supplementary Fig. 4). Singletons are sometimes removed due to possible problems with sequencing errors associated with Illumina and other next-generation sequencing platforms[19]. Removing all OTUs <100 sequences shows the robustness of the pattern and, thus, the apparent prevalence of high diversity and low diversity hosts in coral reef habitat.

Evenness was also high in biotopes with the highest richness and was lowest in the flatworm and nudibranch gut biotopes. Evenness was particularly low in prokaryote communities of the soft coral *Cladiella* sp. (Fig. 2w). For example, 95.5 ± 2.9% (mean ± standard deviation; $n = 4$) of the prokaryote community of *Cladiella* sp. consisted of just three OTUs (OTUs 4, 14 and 17).

**Compositionally distinct but overlapping communities**. There was a highly significant compositional difference among biotopes (Adonis test: $F_{13, 201} = 6.64$, $R^2 = 0.293$, $P < 0.001$; Fig. 3a). The factor biotope, thus, explained almost 30% of the variation in

OTU composition. The main axis of variation (axis 1) separated samples of algae, chitons, sediment, stony corals and the sea cucumber gut and mantle from samples of sponges, sponge denizens, seawater and the nudibranch gut and mantle. Samples from the flatworm, soft coral and sea urchin biotopes were intermediate. The second axis of variation (axis 2 in Fig. 3a) separated a cluster of sponge and seawater samples at high axis 2 values from a cluster of sponge, nudibranch gut and mantle and sponge denizen samples at low axis 2 values. OTUs that significantly discriminated between pairs of biotopes are presented in Fig. 4 and Supplementary Data 3.

The most abundant OTUs observed in the present study were OTUs 1, 2, 9 and 25, all with >30,000 sequence reads. With the exception of OTU-25, the most abundant OTUs were not the most widespread (in terms of their occurrence in samples), but rather were very abundant in selected hosts (Fig. 4). OTU-2, assigned to *Mycoplasma* sp., and with only 92% sequence similarity to an OTU obtained from the oyster *Crassostrea gigas* from Australia (Gb-Acc: JF827444; Supplementary Data 4), was mainly found in the nudibranch species *Halgerda willeyi* (although it was a rare constituent of the sea cucumber gut and mantle and stony coral biotopes). OTU-9, assigned to the Rhodospirillales order, and with 96% sequence similarity to an OTU obtained from seawater in the Northeast subarctic Pacific Ocean (Gb-Acc: HQ672247), was most abundant in the

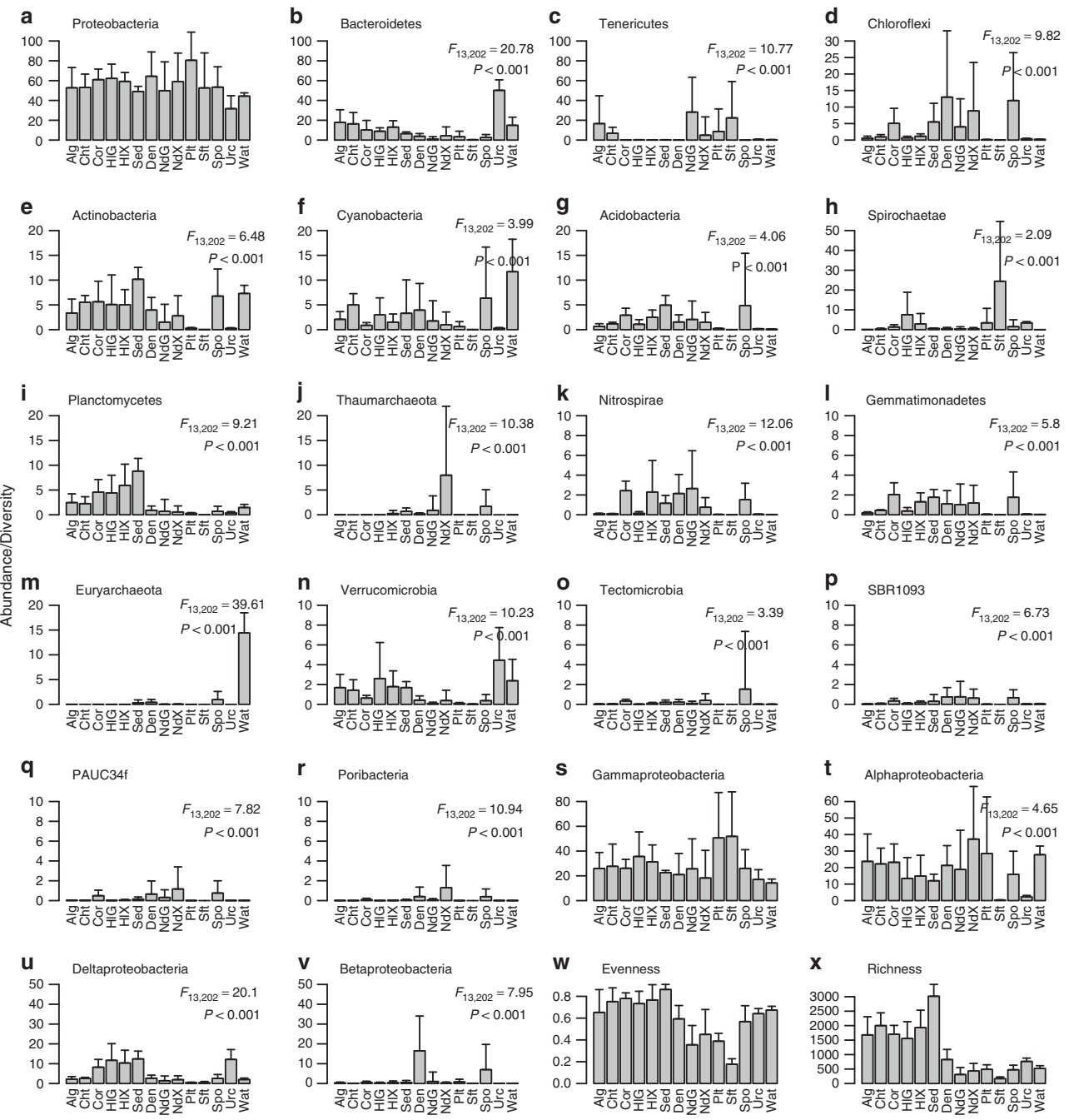

**Fig. 2** Mean relative abundance of the most abundant phyla, proteobacterial classes, OTU richness and evenness. Error bars represent a single standard deviation. **a** Proteobacteria, **b** Bacteroidetes, **c** Tenericutes, **d** Chloroflexi, **e** Actinobacteria, **f** Cyanobacteria, **g** Acidobacteria, **h** Spirochaetae, **i** Planctomycetes, **j** Thaumarchaeota, **k** Nitrospirae, **l** Gemmatimonadetes, **m** Euryarchaeota, **n** Verrucomicrobia, **o** Tectomicrobia, **p** SBR1093, **q** PAUC34f, **r** Poribacteria, **s** Gammaproteobacteria, **t** Alphaproteobacteria, **u** Deltaproteobacteria, **v** Betaproteobacteria and diversity components, **w** Evenness and **x** Richness in the following biotopes: algae (Alg), chitons (Cht), stony corals (Cor), sea cucumber gut (HIG), sea cucumber mantle (HIX), sediment (Sed), sponge denizens (Den), nudibranch gut (NdG), nudibranch mantle (NdX), flatworms (Plt), soft corals (Sft), sponges (Spo), sea urchins (Urc) and seawater (Wat). When significant ($P < 0.0023$; Bonferroni corrected $\alpha$ value), results of the GLM analyses are presented in the top right of the subfigures. Source data are provided as a Source Data file

nudibranch species *Hypselodoris maritima* and *Mexichromis multituberculata*. OTU-1, assigned to the Rhizobiales order, and with 99% sequence similarity to an OTU obtained from the sponge *Tethya californiana* (Gb-Acc: EU290221), was abundant in various *Phyllidia* species. OTU-25, assigned to the genus *Synechococcus*, and with 100% sequence similarity to an OTU obtained from seawater in the Mediterranean Sea (Gb-Acc: MH076976), was the most widespread OTU and was found in

209 (of 216; 96.8% of all samples) samples and was most abundant in seawater samples (Fig. 4).

As can be seen in Fig. 4 and Supplementary Figs. 5, 6 and 7, most of the abundant OTUs, including significantly discriminating OTUs, were recorded in multiple biotopes, albeit oftentimes a rare component of these biotopes. Notable exceptions to this pattern were OTUs assigned to the phylum Tenericutes (e.g. OTU-2), which were highly abundant in selected biotopes and

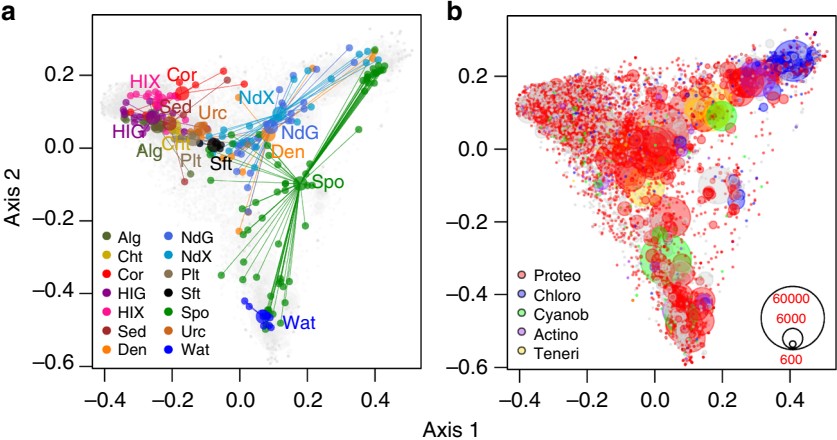

**Fig. 3** Ordination showing the first two axes of the PCO analysis. **a** Symbols represent samples of algae (Alg), chitons (Cht), stony corals (Cor), sea cucumber gut (HIG), sea cucumber mantle (HIX), sediment (Sed), sponge denizens (Den), nudibranch gut (NdG), nudibranch mantle (NdX), flatworms (Plt), soft corals (Sft), sponges (Spo), sea urchins (Urc) and seawater (Wat). Samples from biotopes are connected to group centroids; the figure was produced using the ordispider function in the vegan package. **b** OTU symbols colour-coded according to their taxonomic assignment to selected phyla: Proteobacteria (Proteo), Chloroflexi (Chloro), Cyanobacteria (Cyanob), Actinobacteria (Actino) and Tenericutes (Teneri). The first two axes explain 22% of the variation in the data set. The circle size of the OTU is proportional to their abundance (number of sequences) as indicated by the symbol legend in the bottom right corner of **b**. Source data are provided as a Source Data file

often absent in other biotopes. OTUs found across a range of biotopes included OTUs assigned to phyla that have been deemed to be indicator phyla of HMA sponges, such as Chloroflexi, Acidobacteria and Poribacteria[18,20]. Despite, for example, the relatively high abundance of Chloroflexi in HMA sponges (Fig. 3b and Supplementary Data 1), the most abundant Chloroflexi OTUs were also present in most biotopes, albeit at lower relative abundances (Supplementary Fig. 5). This same pattern held for other abundant phyla, e.g. Acidobacteria and Actinobacteria, but also for less abundant phyla, including Poribacteria, of which OTUs were found in relatively low numbers in a large number of biotopes (Supplementary Fig. 7). In the present study, OTUs assigned to phyla including Chloroflexi, Acidobacteria, Actinobacteria and Poribacteria were present in most biotopes, although they were particularly abundant in HMA sponges, sponge denizens and nudibranchs (Supplementary Data 1).

A large amount of variation in the adonis analysis (~70%) remained unexplained. This is, in part, due to the pronounced overlap among samples from different biotopes or a separation between different groups or species within the same biotope. Within algae, for example, specimens of *Halimeda* sp. were compositionally distinct from other algal species and had lower OTU richness and evenness (Supplementary Data 1). Sponges, in turn, included samples of the species *Acanthella cavernosa*, *Echinodictyum asperum*, *Ptilocaulis spiculifer* and *Stylissa carteri* that clustered with seawater samples (high axis 1 and low axis 2 values; Fig. 4). Species of these genera have been previously identified as low microbial abundance (LMA) sponges[14]. Other sponge samples clustered together with a subset of samples from the sponge denizens and nudibranch gut and mantle biotopes (high axis 1 and high axis 2 values). These were all from the HMA sponges *A. lobata*, *H. erectus* and *X. testudinaria*. Other samples of sponges appeared to house prokaryote communities intermediate in composition between these two previous clusters (high axis 1 and intermediate axis 2 values). These included the agelasids *Agelas nemoechinata* and *Acanthostylotella cornuta*. Finally, a number of sponge samples were compositionally similar to samples from other host taxa with intermediate axis 1 and 2 values (Fig. 3 and Supplementary Data 1). These included samples of *Haliclona cymaeformis*, *Suberites diversicolor* and *Hymeniacidon* sp. (Supplementary Data 1).

**HMA sponges have low richness but high evenness**. In general, there was a positive linear relationship between richness and evenness, among biotopes but also within biotopes (Fig. 5a). This figure also highlights that, although there was a continuous variation in prokaryote OTU richness among samples, there appear to be high and low diversity host species, in addition to host species of intermediate diversity. Species hosting some of the richest prokaryote communities included the sea cucumber *H. hilla* (2260 ± 383 OTUs; mantle; $n = 7$), the chiton *L. japonica* (2001 ± 439 OTUs; $n = 3$) and the alga *Padina* sp. (2099 ± 267 OTUs; $n = 3$). In contrast, some of the least diverse prokaryote communities were found in the soft coral *Cladiella* sp. (170 ± 58 OTUs; $n = 4$) and the gut (218 ± 182 OTUs; $n = 3$) and mantle (311 ± 114 OTUs; $n = 4$) of the nudibranch *P. picta*. Species of intermediate diversity included the sponge *E. asperum* (801 ± 311 OTUs; $n = 3$) and the sea urchin *D. savignyi* (764 ± 113 OTUs; $n = 5$). The large standard deviations in richness values within species, particularly in high diversity hosts, highlights that there was also substantial variation within host species. Certain species also deviated from the general trend of increasing richness and evenness. This was most apparent with species in the 'HMA' cluster, *A. lobata*, *H. erectus* and *X. testudinaria*, in addition to certain nudibranch and sponge denizen samples, that were characterised by relatively low richness, but high evenness (Fig. 5a; encircled in red; Supplementary Data 1).

**Is everything everywhere?** In order to study the distribution of OTUs among biotopes, we created a subset of the total dataset only including OTUs with >100 sequences (hereafter called $OTUs_{100}$; Supplementary Data 5). This subset included 1731 $OTUs_{100}$ and 1,922,781 sequences (89% of all sequences). In this subset, only a very small percentage (1.2%; 21 $OTUs_{100}$) of $OTUs_{100}$ were restricted to a single biotope and less than 3.9% (69 $OTUs_{100}$) were restricted to one or two biotopes (Fig. 5b and Supplementary Data 5). Of the 21 $OTUs_{100}$ restricted to a single biotope, all except three (restricted to the sea cucumber mantle) were only found in sponges. Thirty-four of the 48 $OTUs_{100}$ restricted to two biotopes were also found in sponges and another biotope. An additional 11 were found in the sea cucumber gut and/or mantle biotopes (Supplementary Data 5).

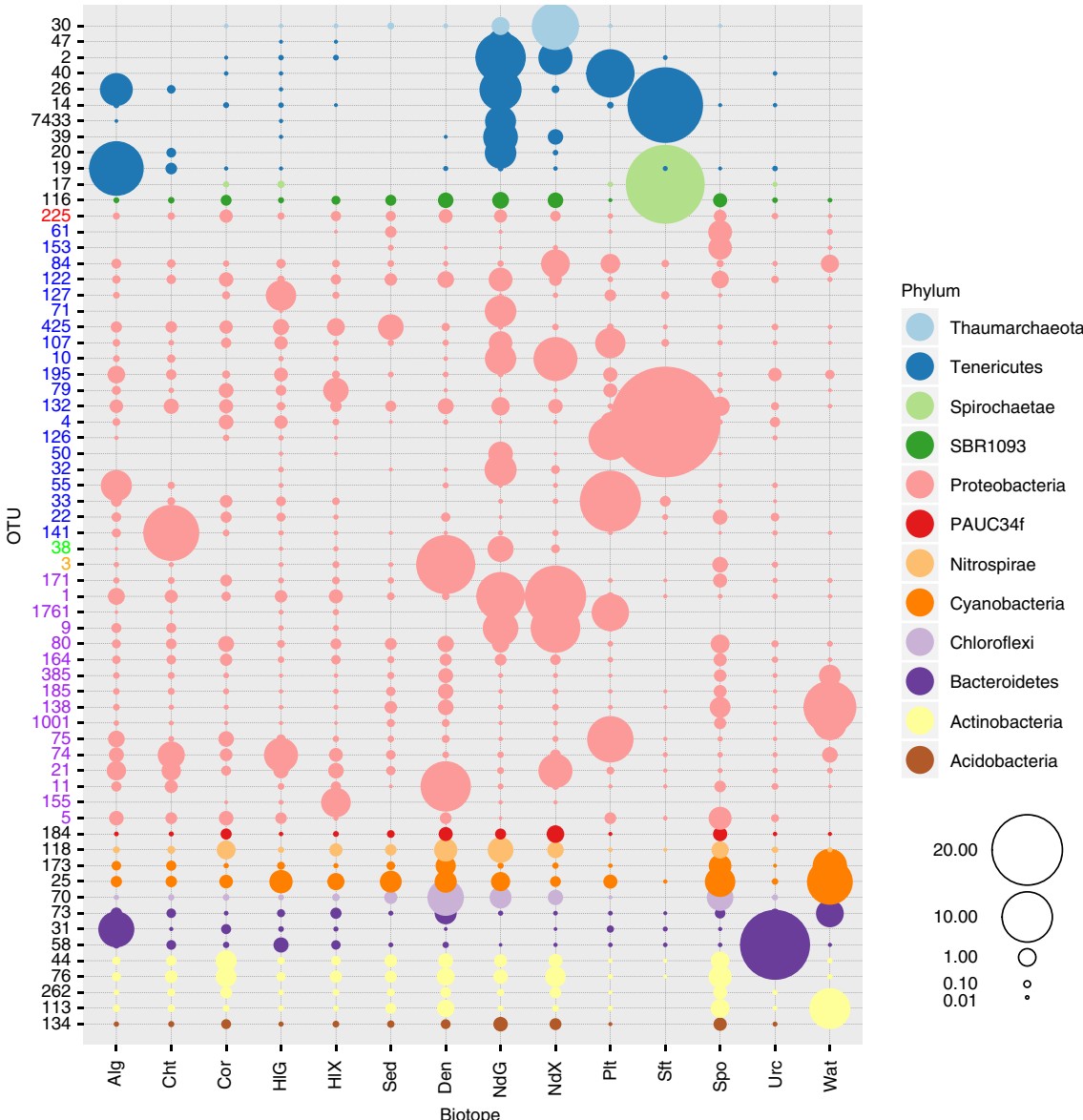

**Fig. 4** Relative abundance of significantly discriminating OTUs ($P < 0.001$) identified using Simper. Symbols are colour-coded according to prokaryote phylum. Codes on the x-axis represent algae (Alg), chitons (Cht), stony corals (Cor), sea cucumber gut (HlG), sea cucumber mantle (HlX), sediment (Sed), sponge denizens (Den), nudibranch gut (NdG), nudibranch mantle (NdX), flatworms (Plt), soft corals (Sft), sponges (Spo), sea urchins (Urc) and seawater (Wat). The circle size of the OTU is proportional to the mean percentage of sequences per biotope as indicated by the symbol legend in the bottom right corner of the figure. The y-axis shows the OTU id number. The y-axis numbers have been colour coded for the proteobacterial OTUs to identify class assignment; red: JTB23, blue: Gammaproteobacteria, green: Epsilonproteobacteria, orange: Betaproteobacteria and purple: Alphaproteobacteria. Source data are provided as a Source Data file

The 21 OTUs$_{100}$ restricted to a single biotope, give us a new look into the rare members of the coral reef prokaryote metacommunity. The total abundance of those OTUs$_{100}$ varied from 102 (0.005% of OTUs$_{100}$ sequences) to 905 (0.11%) sequences (Supplementary Data 5). The most abundant of these (OTUs-579) was restricted to sponges and assigned to the Latescibacteria phylum with only 81% sequence similarity to an organism previously obtained from a deep-sea octocoral (Gb-Acc: DQ395794). The most abundant OTUs$_{100}$ restricted to two biotopes included OTUs 71, 550 and 762. OTUs 550 and 762 were restricted to the sponge and sediment biotopes while OTU-71 was restricted to the sea cucumber and nudibranch gut biotopes. OTU-71, assigned to the gammaproteobacterial order HTA4, had 92% sequence similarity to an organism obtained from black deposit in a lava tube from a cave in the Canary

Islands (Gb-Acc: LT702969). OTU-550, assigned to the Caldilineaceae (Chloroflexi), had 95% sequence similarity with an organism obtained from the sponge *Agelas dilatata* (Gb-Acc: EF076192). OTU-762, assigned to the Gemmatimonadetes, had 98% sequence similarity with an organism obtained from the sponge *Amphimedon compressa* (Gb-Acc: GU984210).

In Fig. 5c, it can be seen that there is both a wide variation in the number of OTUs$_{100}$ found in a single biotope, and a rapid increase in the number of total OTUs$_{100}$ sampled as biotopes are added. To explore this further, we assessed the number of OTUs$_{100}$ shared among biotopes (Fig. 5d). Figure 5d shows the numbers of OTUs$_{100}$ shared among five biotopes, namely, sediment, the sea cucumber gut, algae, sponge and nudibranch mantle biotopes. All but 4 OTUs$_{100}$ (99.8% of all OTUs$_{100}$; Fig. 5d) were found in these five biotopes. These five biotopes

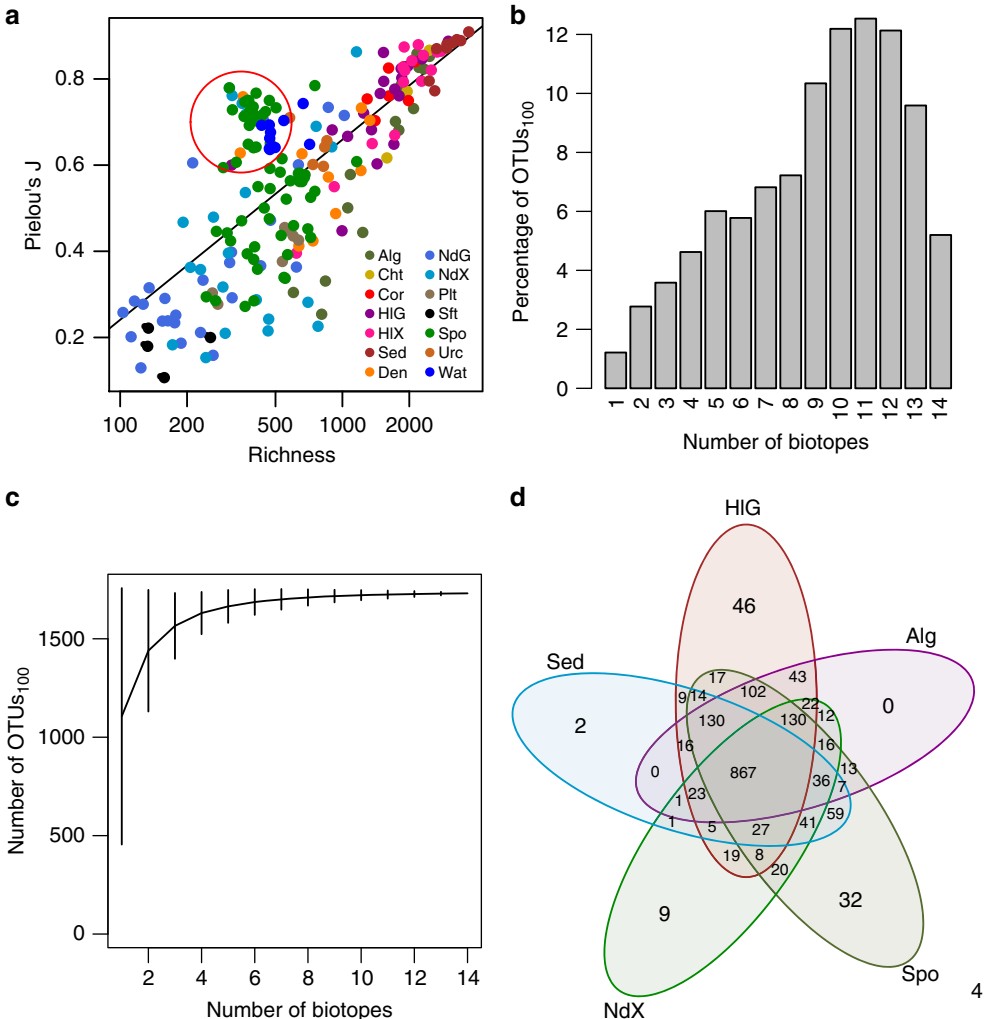

**Fig. 5** Diversity components and distribution of OTUs among biotopes. **a** Relationship between richness and evenness. OTUs representing HMA sponges have been encircled in red. **b** Percentage of $OTUs_{100}$ recorded in from 1 to 14 biotopes. For example, 1.2% of $OTUs_{100}$ (21 $OTUs_{100}$) were recorded in one biotope, 2.8% (48 $OTUs_{100}$) in two biotopes, 3.6% (62 $OTUs_{100}$) in three biotopes and 5.2% (90 $OTUs_{100}$) in all 14 of the main biotopes. **c** Rarefied OTU richness (error bars represent 95% confidence intervals) as a function of the number of biotopes sampled and estimated using the specaccum function in vegan with the 'method' argument set to 'random' and 999 permutations. **d** Venn diagram, obtained using the vennCounts and vennDiagram functions of the limma package in R, showing the number of OTUs shared among the following five biotopes: algae (Alg), holothurian gut (HIG), sponges (Spo), sediment (Sed) and nudibranch mantle (NdX). Source data are provided as a Source Data file

shared 867 $OTUs_{100}$, while 2 $OTUs_{100}$ were only found in sediment, 0 in algae, 9 in the nudibranch mantle, 32 in sponges and 46 in the sea cucumber gut. Note that these $OTUs_{100}$ may be present in other biotopes. Sponges shared 59 $OTUs_{100}$ with sediment, which were not shared with the other biotopes compared to 9 $OTUs_{100}$ shared between sediment and the sea cucumber gut. Note that just three biotopes, namely, the sea cucumber gut, sponges and nudibranch mantle encompassed all but 6 $OTUs_{100}$ (99.7% of all $OTUs_{100}$).

**Environmental OTUs in host-associated prokaryote communities.** In order to study the influence of seawater, sediment and the broader surrounding environment (sediment and seawater) on prokaryote composition in our host biotopes, we assessed the number and percentage of $OTUs_{100}$ in each host that were also found in (1) sediment but not seawater (hereafter known as sediment $OTUs_{100}$), (2) seawater but not sediment (hereafter known as seawater $OTUs_{100}$) and (3) sediment and/or seawater (hereafter known as environmental $OTUs_{100}$). Note that category 3 (sediment and/or seawater) also includes all $OTUs_{100}$ of categories 1 and 2.

Significantly more sediment $OTUs_{100}$ were recorded in algae, chitons, stony corals and the sea cucumber gut and mantle than all other biotopes with the exception of the sponge denizen and sea urchin biotopes (Fig. 6a and Supplementary Data 2). This also held as a percentage of total $OTUs_{100}$ (Fig. 6d). The number of seawater $OTUs_{100}$ was highest in the algae and chiton biotopes and significantly more so than in the nudibranch gut and mantle, sponge and soft coral biotopes (Fig. 6b and Supplementary Data 2). Algae, chitons, stony corals and the sea cucumber gut and mantle also housed significantly more environmental $OTUs_{100}$ than the nudibranch gut and mantle, flatworm, soft coral and sponge biotopes (Fig. 6c). However, the percentage of environmental $OTUs_{100}$ was significantly higher in sponges than all other biotopes, except chitons (Fig. 6f). Sponges housed a mean of $93.8 \pm 3.5\%$ (representing $91.1 \pm 18.3\%$ of $OTUs_{100}$ sequences; Fig. 6l; $n = 63$) environmental $OTUs_{100}$ compared, for example, to just $71.0 \pm 5.2\%$ ($50.8 \pm 37.2\%$ of $OTUs_{100}$ sequences; $n = 4$) for soft corals, $74.2 \pm 3.5\%$ ($62.9 \pm 12.7\%$ of $OTUs_{100}$ sequences; $n = 5$) for sea urchins and $79.8 \pm 3.9\%$ ($24.6 \pm 18.9\%$ of $OTUs_{100}$ sequences; $n = 7$) for flatworms. Four

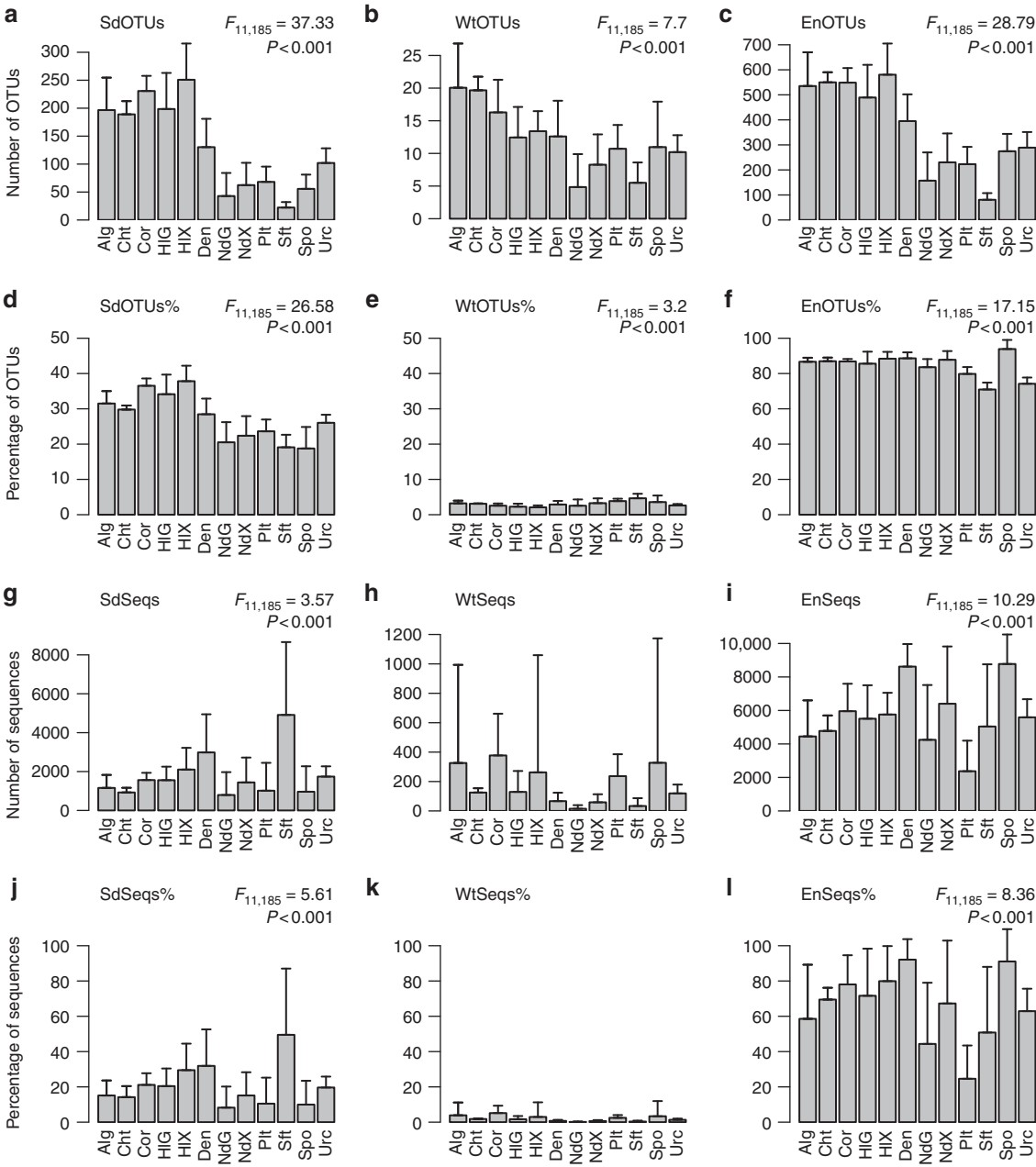

**Fig. 6** Mean counts and percentages of sediment, seawater and environmental OTUs in selected hosts. Error bars represent a single standard deviation. Codes on the *x*-axis represent algae (Alg), chitons (Cht), stony corals (Cor), sea cucumber gut (HIG), sea cucumber mantle (HIX), sponge denizens (Den), nudibranch gut (NdG), nudibranch mantle (NdX), flatworms (Plt), soft corals (Sft), sponges (Spo) and sea urchins (Urc). **a** Number of $OTUs_{100}$ shared with sediment only (SdOTUs), **b** number of $OTUs_{100}$ shared with seawater only (WtOTUs), **c** number of $OTUs_{100}$ shared with sediment and/or seawater (EnOTUs), **d** percentage of $OTUs_{100}$ shared with sediment only (SdOTUs%), **e** percentage of $OTUs_{100}$ shared with seawater only (WtOTUs%), **f** percentage of $OTUs_{100}$ shared with sediment and/or seawater (EnOTUs%), **g** number of sequences shared with sediment only (SdSeqs), **h** number of sequences shared with seawater only (WtSeqs), **i** number of sequences shared with sediment and/or seawater only (EnSeqs), **j** percentage of sequences shared with sediment only (SdSeqs%), **k** percentage of sequences shared with seawater only (WtSeqs%) and **l** percentage of sequences shared with sediment and/or seawater (EnSeqs%). Results of the GLM analyses are presented in the top right of the subfigures when significant. Source data are provided as a Source Data file

of the most abundant OTUs in flatworms (OTUs 33, 40, 126 and 1761) and two in soft corals (OTUs 14 and 17) were only found in host-associated biotopes and were not found in seawater or sediment, explaining the low percentages of environmental sequences in both biotopes (Fig. 4).

Although, on average, almost 94% of the $OTUs_{100}$ recorded in sponges were found in the surrounding environment (whether sediment or seawater), there was pronounced variation among

sponge species. More than 97% of the $OTUs_{100}$ of *E. asperum*, and *S. carteri* were present in the surrounding environment compared to just $79.7 \pm 5.2\%$ of *H. cymaeformis* ($n = 4$), $86.0 \pm 0.3\%$ of *Paratetilla* sp. ($n = 2$) and $86.6 \pm 2.6\%$ of *Hymeniacidon* sp. ($n = 4$) $OTUs_{100}$. For the HMA sponges, $95.8 \pm 1.9\%$ of *A. lobata* ($n = 2$), $96.3 \pm 1.1\%$ of *H. erectus* ($n = 9$) and $91.5 \pm 1.1\%$ of *X. testudinaria* ($n = 9$) $OTUs_{100}$ were found in the surrounding environment. The very high prevalence of 'environmental' OTUs

in certain sponge species would appear to support the prevalence of horizontal transmission in sponge-prokaryote dynamics. However, sponges may also seed the abiotic environment with their prokaryote symbionts.

## Discussion

The present study revealed pronounced differences in composition and diversity among host-associated biotopes. The great majority of OTUs$_{100}$, however, were recorded in multiple biotopes and a large percentage of OTUs were shared with environmental samples (sediment and/or seawater) with the highest percentage found in sponges. Despite the prevalence of environmental OTUs in sponges, there was pronounced compositional variation between sponges and other host taxa and among sponge species. Certain species, for example, housed prokaryote communities similar to seawater (LMA sponges) while others (HMA sponges) housed communities similar to those found in certain samples of nudibranchs and sponge denizens.

A number of studies have previously remarked on the greater compositional similarity of the prokaryote communities of HMA as opposed to LMA sponges and the greater prevalence of transient (seawater) bacteria in the latter[21–24]. LMA sponges have also been shown to be dominated by different sets of highly abundant OTUs and sometimes even a single dominant OTU[8,15,16,25–28]. Compare this to the prokaryote communities of the HMA sponge species X. testudinaria where the core community of 44 specimens sampled across the vast expanse of the Indo-Pacific region consisted of 71 OTUs representing 57.5% of sequences on average[29].

The greater evenness of HMA sponge species observed in the present study and other studies may help to explain the greater similarity and limited prevalence of transient bacteria in these sponges[23]. Importantly, species evenness has been shown to be positively related to invasion resistance, presumably by limiting the invaders access to available resources[30]. The question remains, however, as to why HMA sponges house more even (and compositionally similar) prokaryote communities than LMA sponges. Previous studies have shown that certain sponge species are able to transmit microorganisms through their larvae (vertical transmission) and suggested that this plays an important role in structuring the prokaryote community[3,31,32]. Other studies have focused on horizontal transmission, e.g. from water column to sponge[33] and the ability of sponges to selectively recruit specific microbial symbionts from seawater[3,12,34]. The actual degree to which the sponge prokaryote community is shaped by both forms of transmission, however, remains largely unknown.

The compositional similarity between certain sponge samples and samples of nudibranchs and sponge denizens suggests that sponges may influence the prokaryote composition of organisms that live on or within them or that feed on them. The sponge denizen biotope included sea cucumbers and barnacles that lived within or on the sponge, presumably for much of their life[35,36]. Nudibranchs, however, are more mobile and may represent vectors carrying microorganisms from one sponge to the other. Nudibranchs also come into intimate contact with their sponge prey during feeding whereby certain species evert and extend their pharyngeal bulb deep into the sponge[37]. During this process, they are also able to sequester toxins from the sponge for their own defence[38]. Our results indicated that the gut and/or mantle prokaryote communities of specimens from certain nudibranch species (Doriprismatica atromarginata, Phyllidiella pustulosa, Phyllidiella nigra, Phyllidia ocellata and Phyllidia elegans) closely resembled that of sponge prokaryote communities. All of these nudibranch species have been recorded feeding on sponges[37,39–41]. A number of these specimens were also collected from sponges while diving. Specimens of the sea cucumber Synaptula sp., a

sponge denizen sampled from X. testudinaria, housed a prokaryote community similar to that of the 'HMA' sponge cluster, which included X. testudinaria. Members of the genus Synaptula are often common in coral reef habitat, particularly in association with sponges and can sometimes be so abundant that they cover the sponge's surface. They have also been shown to be able to exploit sponge exudates[35]. Interestingly, the barnacle Acasta sp., which was collected within X. testudinaria, was the only sponge denizen barnacle that also housed a prokaryote community similar to that of members of the 'HMA' sponge cluster. The other sponge barnacles were collected within samples of the sponge species Dasychalina fragilis, Agelas cavernosa and Cinachyrella sp.

The similarity between the prokaryote communities of sponges and the guts of certain nudibranch samples, may be an indication that the nudibranch gut communities are dominated by transient microorganisms derived from their preferred food source, namely sponges[37,38]. An individual's diet can have a profound effect on gut prokaryote composition[42,43]. This difference can extend to species, whereby there are marked differences in gut microbiome composition among mammal species with different diets[44,45]. This distinction appears to apply to nudibranchs, whereby the gut and mantle prokaryote communities of species known to feed on sponges closely resembled that of certain sponge species (Fig. 3). It would be interesting to test how different diets (e.g. different sponge species) affect the nudibranch prokaryote community.

The very high number of OTUs shared among different biotopes would appear to lend support to the 'everything is everywhere but the environment selects' hypothesis of Baas Becking[46]. In line with this, the very high richness and evenness of sediment suggest that it may function as a microbial seed bank. There was also considerable compositional similarity between sediment samples and high diversity host samples of algae, stony corals and sea cucumbers among others. In contrast, seawater samples were only compositionally similar to samples of certain sponge species. Previously, Cleary and Polónia[47] also showed that populations of mussels inhabiting Indonesian marine lakes and mangroves shared much more OTUs with sediment than with seawater and were compositionally more similar to sediment than to seawater. Gibbons et al.[48] previously suggested that the marine biosphere maintains a persistent microbial seed bank. In their scenario, all microbes are found everywhere due to the immensity and persistence of this seed bank, and apparent local or host-associated endemism is merely a result of insufficient sequencing. Community structure is, thus, a function of relative abundance rather than the presence or absence of certain microbial taxa. The presence of such a seed bank has repercussions for ecological theory, given the limited importance of long-distance dispersal and the ability of low abundance populations to rapidly expand when the appropriate environment is encountered[48]. In the global marine environment, hydrographic parameters of seawater masses, furthermore, greatly contribute to the dispersion of sediment microbial communities at regional and global scales, although microbial cell dispersal is highly dependent on the ability to tolerate stress[49,50].

Although wide in scope, the present study only represents a small fraction of marine species in the coral reef environment and even in this dataset, there was considerable variation among species within biotopes. Much more research is needed to understand the variation in microbial composition of taxa such as sea cucumbers, flatworms, algae and nudibranchs. A large amount of time and resources have been spent studying the prokaryote communities of a limited number of taxa leaving large gaps in our knowledge of the coral reef metacommunity. Sponges have been deemed major contributors to total microbial diversity in the world's oceans, and are considered to be reservoirs of exceptional microbial diversity[2] without, however, having actually

studied other host taxa in detail. In coral reefs, sponges do not appear to stand alone as the main contributors to total prokaryote diversity as this study highlights; other biotopes host more diverse prokaryote communities, e.g. sea cucumbers. The present study shows that sponges are only one, albeit an interesting, component of a much larger coral reef metacommunity.

## Methods

**Sampling locations.** All host-associated, sediment and seawater samples were collected from various sites in Taiwan and Thailand (Supplementary Data 1). All locations were coral reef habitat. A detailed description of the Taiwanese sampling sites can be found in Coelho et al.[51] and Huang et al.[52] and meta data for all samples including the sampling location and time of sampling can be found in Supplementary Data 1. Fragments of host individuals were collected using SCUBA diving, or snorkelling, including the surface and interior or the whole organism (depending on the size) in order to sample as much as possible of the whole prokaryote community. Sediment was collected from the upper 5 cm surface layer using a plastic disposable syringe from which the end had been cut in order to facilitate sampling. Seawater was collected between the depths of 1–2 m with a 1.5 L bottle and subsequently 1 L (±50 ml) of water was filtered through a Millipore® White Isopore Membrane Filter (0.22 μm pore size) to obtain seawater prokaryote communities. All samples were subsequently preserved in 96% EtOH. All samples were kept cool (<4 °C) immediately after collection and during transport. In the laboratory, samples were stored at −20 °C until DNA extraction.

A total of 216 samples belonging to algae, chitons, stony corals, sea cucumbers, sponge denizens (organisms that live on or within sponges), nudibranchs, flatworms, soft corals, sponges, sea urchins, water and sediment were collected. In the present study, all samples were assigned to 14 biotopes, which included the guts and mantles of sea cucumbers and nudibranchs as separate biotopes. Certain biotopes were well represented, e.g. sponges (63 samples from 18 species) and nudibranchs (48 samples from 13 species) while others only consisted of a just few samples and/or a single species., e.g. soft corals (4 samples from the species *Cladiella* sp.), chitons (3 samples from the species *L. japonica*) and sea urchins (5 samples from the species *D. savignyi*). All the samples used in the present study can be found in Supplementary Data 1 including the sampling site and taxonomic identification.

**DNA extraction and next-generation sequencing analysis.** PCR-ready genomic DNA was isolated from all samples using the FastDNA® SPIN soil Kit (MPbiomedicals) following the manufacturer's instructions. Briefly, the whole membrane filter (for seawater samples) and ±500 mg of sediment and host specimens (including parts of the surface and/or interior) were cut into small pieces (in the case of the membrane filter and host specimens) and transferred to Lysing Matrix E tubes containing a mixture of ceramic and silica particles. A blank control, in which no tissue was added to the Lysing Matrix E tubes, was also included. The microbial cell lysis was performed in the FastPrep® Instrument (Q Biogene) for 80 s at 6.0 ms$^{-1}$. The extracted DNA was eluted into DNase/Pyrogen-Free Water to a final volume of 50 μl and stored at −20 °C until use. The 16S rRNA gene V3V4 variable region PCR primers 341F 5′-CCTACGGGNGGCWGCAG-3′ and 785R 5′-GACTACHVGGGTATC-TAATCC-3′ [53] with barcode on the forward primer were used in a 30 cycle PCR assay using the HotStarTaq Plus Master Mix Kit (Qiagen, USA) under the following conditions: 94 °C for 3 min, followed by 28 cycles of 94 °C for 30 s, 53 °C for 40 s and 72 °C for 1 min, after which a final elongation step at 72 °C for 5 min was performed. After amplification, PCR products were checked in 2% agarose gel to determine the success of amplification and the relative intensity of bands; the blank control did not yield any bands. Multiple samples were pooled together in equal proportions based on their molecular weight and DNA concentrations. Pooled samples were purified using calibrated Ampure XP beads. Pooled and purified PCR product was used to prepare the DNA library following the Illumina TruSeq DNA library preparation protocol. Next-generation, paired-end sequencing was performed at MrDNA (Molecular Research LP; http://www.mrdnalab.com/; last checked 18 November 2016) on an Illumina MiSeq device (Illumina Inc., San Diego, CA, USA) following the manufacturer's guidelines. Sequences from each end were joined following Q25 quality trimming of the ends followed by reorienting any 3′–5′ reads back into 5′–3′ and removal of short reads (<150 bp). The resultant files were analysed using the QIIME (Quantitative Insights Into Microbial Ecology)[54] software package (http://www.qiime.org/) and USEARCH10 [19].

**16S rRNA gene sequencing analysis.** For a detailed description of the sequence analysis, see Coelho et al.[51] and Cleary et al.[55]. Briefly, in QIIME, fasta and qual files were used as input for the split_libraries.py script in QIIME. Default arguments were used except for the minimum sequence length, which was set at 250 base pairs (bps) after removal of forward primers and barcodes. Using USEARCH10 (https://www.drive5.com/usearch/; last checked 2019 02 11), reads were filtered with the -fastq_filter command and the following arguments: -fastq_trunclen 250 -fastq_maxee 0.5 -fastq_truncqual 15. Sequences were then dereplicated and sorted using the -derep_fulllength and -sortbysize commands. OTU clustering (97% sequence similarity threshold) was performed using the -cluster_otus command of USEARCH10 yielding 12025383 sequences assigned to

48880 OTUs. Potential contaminants were removed from the OTU table if they occurred at least two times in the blank control. This conservative measure was chosen because of observations of bleeding between samples from Illumina sequencing and the appearance of abundant reads in blank controls with very low counts[56,57]. Based on this procedure, 958995 sequences and 77 OTUs were removed from the non-rarefied OTU table. OTUs not classified as Bacteria or Archaea or classified as chloroplasts and mitochondria were also removed. Taxonomy was assigned to reference sequences of OTUs using default arguments in the assign_taxonomy.py script in QIIME using the SILVA_128_QIIME_release database and the uclust classifier method[58]. The make_otu_table.py script in QIIME was used to generate a square matrix of OTUs × SAMPLES and subsequently rarefied to 10,000 sequences per sample with the single_rarefaction.py script in QIIME yielding 2,160,000 sequences and 30,725 OTUs. This rarefied table was used as input for further analyses using the R language for statistical computing and has been included as a source data file (https://www.r-project.org/; last checked 2018–07–17).

**Statistical analysis.** A data matrix containing OTU counts per sample was imported into R using the read.csv() function. This table was used to compare community composition, estimate richness and assess the relative abundance of selected higher taxa and is included as a Source Data file. The OTU abundance matrix was $\log_e (x+1)$ transformed (in order to normalise the distribution of the data) and a distance matrix constructed using the Bray–Curtis index with the vegdist() function in the vegan package[59]. The Bray–Curtis index is one of the most frequently applied (dis)similarity indices used in ecology[60–63]. Variation in prokaryote composition among biotopes was assessed with Principal Coordinates Analysis (PCO) using the cmdscale() function in R with the Bray–Curtis distance matrix as input. Variation among biotopes was tested for significance using the adonis() function in vegan. In the adonis analysis, the Bray–Curtis distance matrix of species composition was the response variable with biotope as independent variable. The number of permutations was set at 999; all other arguments used the default values set in the function. Weighted average scores were computed for OTUs on the first four PCO axes using the wascores() function in the vegan package. The simper() function in vegan was used to identify significantly discriminating OTUs between pairs of biotopes based on the $\log_e (x+1)$ transformed OTU table and 999 permutations. The discriminating OTUs contribute the most to differences between pairs of biotopes.

We tested for significant differences in the relative abundance of 18 of the most abundant phyla, the four most abundant proteobacterial classes, and the count and relative abundance of sediment and environmental OTUs among biotopes with an analysis of deviance using the glm() function in R. For the most abundant phyla, proteobacterial classes, and the relative abundance of sediment and environmental OTUs, we first applied a generalized linear model (GLM) with the family argument set to binomial. The ratio, however, of residual deviance to residual d.f. in the models substantially exceeded 1 so we set family to 'quasibinomial'. In the 'quasibinomial' family, the dispersion parameter is not fixed at one so that it can model over-dispersion. For the counts of sediment and environmental OTUs, we set the family argument to 'quasipoisson'. For the least abundant phyla and the two least abundant proteobacterial classes, which included zero counts in the samples, we set the family argument to 'tweedie'[64] with var.power = 1.5 and link.power = 0 (a compound Poisson–gamma distribution). Using the glm models, we tested for significant variation among biotopes using the anova() function in R with the $F$ test, which is most appropriate when dispersion is estimated by moments as is the case with quasibinomial fits. We subsequently used the emmeans() function in the emmeans library[65] to perform multiple comparisons of mean abundance among biotopes using the false discovery rate (fdr) method in the adjust argument. Additional graphs were produced using the ggplot[66] and limma[67] packages. Detailed descriptions of the functions used here can be found in R (e.g.?cmdscale) and online in reference manuals (http://cran.r-project.org/web/packages/vegan/index.html).

**Reporting summary.** Further information on experimental design is available in the Nature Research Reporting Summary linked to this article.

## Data availability
The DNA sequences generated in this study can be downloaded from NCBI BioProject IDs: PRJNA382576, PRJNA397173, PRJNA397177 and PRJNA397178. The source data underlying Figs. 2–6 and Supplementary Figs. 1–7 are provided as a Source Data file.

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

## Acknowledgements

This work was supported by European Funds through COMPETE [FCOMP-01-0124-FEDER-008657] and by National Funds through the Portuguese Foundation for Science and Technology (FCT) within the LESS CORAL [PTDC/AAC-AMB/115304/2009] and Ecotech-Sponge (PTDC/BIAMIC/6473/2014 – POCI-01-0145-FEDER-016531) projects. This work is also part of the research programmes NWO-VIDI with project number 16.161.301 and ASPASIA (015.010.030), which are both (partly) financed by the Netherlands Organisation for Scientific Research (NWO). Thanks are due for financial support to CESAM (UID/AMB/50017 - POCI-01-0145-FEDER-007638), to FCT/MCTES through national funds (PIDDAC), and co-funding by FEDER, within the PT2020 Partnership Agreement and Compete 2020. Francisco J.R.C. Coelho (postdoctoral scholarship: SFRH/BPD/92366/2013), Ana R.M. Polónia (postdoctoral scholarship: SFRH/BPD/117563/2016) and Marina R.S. Ferreira (Ph.D. scholarship: SFRH/BD/114809/2016), were supported by scholarships funded by FCT, Portugal within the Human Capital Operational Programme (HCOP), subsidised by the European Social Fund (ESF) and national funds (MCTES). This research was facilitated by the generous support of the Ministry of Science and Technology (MOST) and the Marine National Parks Headquarter (MNPH), Taiwan to Y.M.H. (MOST 105-2621-B-346-002 and MNPH 104403). All specimens collected in Taiwan were under a permit (No. 20160316364) issued by MNPH. The authors would also like to acknowledge Ana Cecilia Pires, Helder Gomes, Bastian Reijnen and Niels van der Windt for their support in the laboratory and Devrim Gunsel Zahir, Chad Scott, Gregory Hanigan, Floris Cleary, Julian Cleary, Yi-Chin Wu, Yuan-Hao Lin, Ming-Hong Chang, Miao-Yin Syu and You-Hua Lin for their help in the field.

## Author contributions

Fieldwork: D.F.R.C., N.J.d.V., Y.M.H. and S.P. Labwork: T.S., E.v.d.E., M.R.S.F., A.R.M.P., D.F.R.C. and N.C.M.G. Data analysis: D.F.R.C., F.J.R.C.C. and L.C. Writing: D.F.R.C., N.J.d.V., Y.M.H., S.P., T.S., E.v.d.E., M.R.S.F., A.R.M.P., N.C.M.G., F.J.R.C.C., L.C. and J.-P.U.

## Additional information

**Competing interests:** The authors declare no competing interests.

