## [Peer Review File · Nature Communications]

Reviewers' Comments:

Reviewer #1:

Remarks to the Author:

The manuscript titled 'The coral reef microbiome, is everything everywhere after all?' by Daniel Cleary et al. investigate bacterial communities associated with wide range of biotic and abiotic biotopes in coral reefs from Taiwan (Penghu islands) and Thailand (various locations). The authors address an interesting topic that is the metacommunity theory to explain how local bacterial communities are affecting each other through dispersal or flow in coral reefs. The paper is focused on the specificity of sponge-associated microbes through comparison to other microbial-host associations in coral reefs. One of the main findings of the paper is that biotypes have many overlapping bacterial groups, even though they harbor very different bacterial community structures. The authors also find positive correlations between high evenness and high richness except for HMA sponges and sponge denizens. Nudibranch guts and sponge denizen resembled sponges' bacterial communities, and the authors suggested them as vectors carrying bacteria from one sponge to the other. Lastly, the authors give an extensive description of what OTUs are shared between sponges and other biotypes, concluding that everything is everywhere, in agreement with Becking (1934), and that local or host-associated endemism is likely the result of insufficient sequencing.

Major comments

1. Title and aim of the paper:

While I recognize the vast range of coral reef biotypes sampled, it should be made explicit that the paper has a limited interest in approaching the data from a coral reef perspective; rather all analyses and interpretations were restricted to explaining sponges-associated microbiomes as components of the coral reef metacommunity. This is also reflected in the sample set (Supplementary Table 1): only 2 species of stony corals were sampled, and the majority of all species were sampled with only 3 replicates; in contrast, sponge specimens were extensively sampled. This in itself creates a bias with regard to analysis and results obtained, and this should be made explicit. For this reason, the title should be toned down and specifically state that this study's focus are sponges and how their microbial community relates to other coral reef organisms. As the title currently reads I would have expected to see discussed what common patterns of bacterial diversity and potential fluxes of species explain the main clusters formed in your ordination analysis (One formed by Sed, Cor, HIG, HIX, Alg, and another formed by NdG, NdX, Den, and HMA). It would have also been interesting to see what bacterial groups are shared between each biotype and the "background community" (sediments and water) to infer how many of these groups are transient and how many colonize. With regard to the first sentence of the abstract, I would argue that much recent marine microbial research has focused on the coral microbiome, which to me is diagnostic of the rest of the paper. It is written in a sponge centric view, which is fine, but this needs to be stated and conclusions/results have to be re-phrased accordingly.

2. The relevance of the results

The results are somewhat predictable and not completely novel. Most of the current results are descriptive, regularly to contrast bacterial communities of HMA and LMA against the rest of biotypes, however, no concrete patterns contributing to metacommunities were found. Future efforts should target a larger geographic area in order to expand the significance of the current observations. Also, while I am personally a fan of HMA and LMA, I found that this is not a universally accepted concept by the sponge community. The authors should better highlight that this is a controversial concept and highlight the pros and cons.

3. Graphical representations

In some cases, figures contain excessive amounts and not necessarily essential data. For instance:

Figure 2 and 3: Single bars are useful to contrast the abundance of particular taxa among samples. However, I found it very hard to find general patterns from this type of representation, especially when samples are split into two plots. I personally would prefer to see also a single plot with stacked bars representing the means per biotype.

Figure 4a-b: You could increase the area of plotting or reduce the font and circle size to better appreciate the cluster of biotypes on the lower left corner (Sed, Cor, HIG, HIX, Alg)

Figure 4a: I don't see the need in plotting different grey circle sizes to represent OTUs abundance, it's rather confusing.

Figure 4b: Does circle size represent OTUs abundance?

Figure 5 and 6: I am hesitant about how these figures contribute to the line of the paper. If you are looking for indicator species, you shouldn't be restricting the search to the most abundant OTUs. Why didn't you represent e.g. SIMPER results instead? Or if you want to highlight key species, why not showing the abundance of Poribacteria across samples as an example of indicator species dynamics. I would rather send figures 5 and 6 to the supplementary data along with cutting down the text describing them.

Fig 7b-c need a better explanation. I would have expected to see more of this type of information. The idea of relating the number of OTUs and the biotypes in which OTUs were present is good and help to give some context of what proportion of OTUs are host-specific or shared.

Other comments

I would have liked to see how many sequences per sample before and after quality check, as well as how many OTUs per sample before and after contamination removal.

Figure 5: Please state that x-axis is showing OTU ID

L88-91: Does "Sequences" refer to bacteria taxa or genes? Please specify.

L157: Replace "split" with "clustered"

L500: Add primer reference (Klindworth et al. 2013)

L522, L525-26: I would combine the two phrases "OTUs were selected using USEARCH10 (Edgar 2013)" and "OTU clustering (97% sequence similarity threshold) was performed using the -cluster_otus command" into something similar to: "OTU clustering (97% sequence similarity threshold) was performed using the -cluster_otus command of USEARCH10 (Edgar 2013)"

L527: Can you specify what is your blank control? E.g., DNA extraction, PCR or sequencing blank

L866: What do you mean by groups? biotypes? Please clarify.

Reviewer #2:

Remarks to the Author:

This paper combines the sequencing of 22 biotopes cohabiting coral reef systems in order to analyse the level of overlap of microbial species (as OTUs) among them. The effort put in this study is valuable for the scientific community, and it is a novel result. Showing that microbial communities are largely shared across biotopes and that microbial communities of sponges are diverse but not the most diverse, as many times suggested, are important outcomes. These concepts can only really be explored with such large dataset as the one contained within this study.

The manuscript is well written and well organised, the methods are clearly explained and the conclusions are straight forward. However, the samples are not well described. Most come from two previous studies in Penghu island, but it is not easy to know which ones belong to either previous work. Moreover, samples are missing important metadata information, for instance, specific location of the sample within the archipelago, year of the sampling, depth of the samples taken, etc, which could be important factors affecting their results.

I also think the analyses are superficial and general, and much more information and patterns could have been extracted from the data, mostly if the metadata is included, or by analysing more closely some groups. For instance, when a biotope shows different compositions (fig.4), there are no further analyses conducted that could explain those differences related to other factors (specific species, location, etc). Only in the case of LMA sponge, the differences were related to specific species, showing a group of species harbouring bacterial communities similar to seawater and other species including communities similar to host-related biotopes. That analysis could be interesting for nudibranchs (or others) as specified bellow. Also knowing that microbial communities can be very variable in temporal and spatial scales, these factors should be considered in any analysis.

For these reasons I have recommended a rejection of the manuscript, but I encourage a resubmission after some extra work is done.

To guide the authors in a resubmission, I have these comments:

Line 116. results and discussion?

Line 159. Change to "also included samples that were"

Line 161. Move 'other' at the end of sentence. And in similar subsequent sentences.

Line 173. What does it mean "most abundant" in here and in Fig 5 and 6? What is the threshold for the OTUs shown?

Lines 192 to 194. I believe this information still relates with figure 5. If so, Chloroflexi lineages are not "absent" in seawater, algae, or LMA sponges. As you say in line 174, chloroflexi seems present in all those but in low abundances. I can see a small dot in those groups. But I can not see is the ascidians biotope in figure 5.

Line 344. Maybe change "However, the similarity" to "This similarity"?

Lines 351 - 356. You suggest that the nudibranch gut bacteria is related to the preferred food, however, I am not sure that there was information about the specific situation of the nudibranch at

the time of collection (i.e. sampled from the surface of a certain sponge species). I agree it would be interesting to see a study about the effects of diet in the nudibranch gut. But for the moment, I wonder if the authors have ruled out the possibility of the different gut communities being related to the nudibranch taxonomy or the sampling location (i.e. Phuket, Penghu, Pattaya, Koh Tao) for instance.

Line 353. I think it refers to Fig 4.

Line 362. Change to "including all OTUs with > 100 sequences"

Line 369. Change to Fig 7d-h

Line 369-383. This section needs rephrasing. Describe each diagram once, with all the information you want to say together.

Line 393. Change 'mean abundance' to 'total abundance' as it refers to the sum of all sequences. Change that in the supplemental table 6 where it says "Abundance" and in the table description says "Sum".

Line 395. Clarify what 'this' refers to in "this also held".

Line 414 - 425. The host-associated endemism (or distribution) of microbial communities refers to bacteria/archaea found in significant numbers in a specific host. Of course, deeper sequencing will show more rare microorganisms being shared with other biotopes, but that does not contradict the apparent host-associated endemism the author refers to. For two reasons:

a) The well-studied host-associated community (of sponges for instance) is a result of evolution and selection in each host species to allow them to survive and grow to significant numbers, not just as 'rare' or transient bacteria.

b) Deeper sequencing will increase the arbitrary threshold of 100 sequences chosen in this work, but what is more important is the percentage they represent to the total sequences. That percentage would not be very affected with deeper sequencing.

So, I think that sentence could be removed, or explained with more convincing arguments (i.e. how deeper sequencing of rare microorganism will contradict the host-associated specific communities).

Line 417. I am assuming you mean 'control' as control for contamination in line 417. The idea of sediments as control is valid, but it seems contradictory to use what seems to be a "seed bank" as control for contamination. If you want to discuss this part better, a more specific analysis including the sediment samples and all the other biotopes showing sediment sequences (not only in groups of 3), and including abundances, would be interesting.

Line 528. We have observed bleeding between samples from Illumina sequencing (reads being incorrectly assigned to samples) and thus are very critical of the QIIME contamination filtering approach. This method removes any OTU found in the blank/control from the dataset. We noticed that our most abundant reads have appeared in our blank controls (with very low counts, but > 0), most likely due to Illumina software/hardware issues, and using the QIIME filtering approach would remove true OTU from your dataset.

Can you please comment on the number of reads and number of OTUs removed in this study, and clarify whether a significant amount of the data was removed?

Line 539-540. The use of "presence and abundance" is ambiguous, because in this field of work there are two approaches to analysing the data: presence/absence or abundances (counts). So it is unclear

to the reader. Change to "OTU abundances or OTU counts".

Line 866. Rephrase b). I did not understand that sentence.

Line 869 -874. Maybe change to: "d-h) Venn diagrams of the number of OTUs shared in groups of three biotopes. Abbr.: algae (Alg), holothurian gut (HIG), HMA sponges (HMA), sediment (Sed), stony corals (Cor), LMA sponges (LMA)".

Fig. 1 is not space efficient. The zoomed regions need to be positioned better to make use of the figure area. Also, I would change the dots in the general map by a square that represents the zoomed region.

Fig 4 to 7. Why are only 13 biotopes instead of all 22 shown in these figures? This is not adequately explained in the text or caption.

Fig 2 and fig 3.

- I believe this graph has been divided in two (13 biotopes fig 2 and 10 biotopes in fig3) because the rest of the plots only show the first 13 biotopes. If that is the case, maybe you can move Figure 3 to supplementary material and only keep the main 13 biotopes in the main text, to be consistent with the rest.

- I would order the biotopes as you named them in lines 122-127. Putting the 'Oth' at the end of the figure. And keep that other in all other figures (4 to 7).

- I don't think you need the letters 'a to x'. The names are clear, and you don't need to explain them in the figure legend.

Fig 8. The colour combinations need to be improved. It is hard to associate the darker green to the overlapping of green and purple. Using some common 2 colour combination could work; blue + red = purple, or green + yellow = orange.

Table S1. Even if samples are from previous papers, it would be helpful if you could still add in the table some of the metadata, at least specific location, date of collection and depths, and any other recorded information.

Table S2. I would show the different families within Proteobacteria rather than just the whole phylum.

Table S6. Maybe it would be more informative to show the abundances in each biotype, instead of just presence/absence and a column with 'total abundance'.

Reviewers' comments:

Reviewer #1 (Remarks to the Author):

The manuscript titled 'The coral reef microbiome, is everything everywhere after all?' by Daniel Cleary et al. investigate bacterial communities associated with wide range of biotic and abiotic biotopes in coral reefs from Taiwan (Penghu islands) and Thailand (various locations). The authors address an interesting topic that is the metacommunity theory to explain how local bacterial communities are affecting each other through dispersal or flow in coral reefs. The paper is focused on the specificity of sponge-associated microbes through comparison to other microbial-host associations in coral reefs. One of the main findings of the paper is that biotypes have many overlapping bacterial groups, even though they harbor very different bacterial community structures. The authors also find positive correlations between high evenness and high richness except for HMA sponges and sponge denizens. Nudibranch guts and sponge denizen resembled sponges' bacterial communities, and the authors suggested them as vectors carrying bacteria from one sponge to the other. Lastly, the authors give an extensive description of what OTUs are shared between sponges and other biotypes, concluding that everything is everywhere, in agreement with Becking (1934), and that local or host-associated endemism is likely the result of insufficient sequencing.

Major comments

1. Title and aim of the paper:

While I recognize the vast range of coral reef biotypes sampled, it should be made explicit that the paper has a limited interest in approaching the data from a coral reef perspective; rather all analyses and interpretations were restricted to explaining sponges-associated microbiomes as components of the coral reef metacommunity. This is also reflected in the sample set (Supplementary Table 1): only 2 species of stony corals were sampled, and the majority of all species were sampled with only 3 replicates; in contrast, sponge specimens were extensively sampled. This in itself creates a bias with regard to analysis and results obtained, and this should be made explicit. For this reason, the title should be toned down and specifically state that this study's focus are sponges and how their microbial community relates to other coral reef organisms. As the title currently reads I would have expected to see discussed what common patterns of bacterial diversity and potential fluxes of species explain the main clusters formed in your ordination analysis (One formed by Sed, Cor, HIG, HIX, Alg, and another formed by NdG, NdX, Den, and HMA). It would have also been interesting to see what bacterial groups are shared between each biotype and the "background community" (sediments and water) to infer how many of these groups are transient and how many colonize. With regard to the first sentence of the abstract, I would argue that much recent marine microbial research has focused on the coral microbiome, which to me is diagnostic of the rest of the paper. It is written in a sponge centric view, which is fine, but this needs to be stated and conclusions/results have to be re-phrased accordingly.

A: The discussion in the manuscript has been shifted to focus more on other non-sponge biotopes including the variation within biotopes such as algae, sea cucumbers and nudibranchs. Differences in composition and diversity between the main clusters are also now explicitly addressed. In addition to this, we have also added a new analysis to assess the contribution of sediment OTUs and all environmental (sediment and/or seawater) OTUs to the prokaryote communities of all host-associated biotopes. This analysis has yielded some important insights that have improved the manuscript. The title has also been changed as suggested.

2. The relevance of the results

The results are somewhat predictable and not completely novel. Most of the current results are descriptive, regularly to contrast bacterial communities of HMA and LMA against the rest of biotypes, however, no concrete patterns contributing to metacommunities were found. Future efforts should target a larger geographic area in order to expand the significance of the current observations. Also, while I am personally a fan of HMA and LMA, I found that this is not a universally accepted concept by the sponge community. The authors should better highlight that this is a controversial concept and highlight the pros and cons.

A: In the revised version, sponges are included as a single biotope and there is no a priori separation into HMA and LMA biotopes. Although less focus is placed on the HMA-LMA dichotomy of sponges overall, mention is made of the observed compositional differences between known HMA and LMA sponges. With respect to the results and analyses, more focus has been placed on non-sponge biotopes and differences in diversity and composition between the metacommunities have been highlighted.

3. Graphical representations

In some cases, figures contain excessive amounts and not necessarily essential data. For instance:

Figure 2 and 3: Single bars are useful to contrast the abundance of particular taxa among samples. However, I found it very hard to find general patterns from this type of representation, especially when samples are split into two plots. I personally would prefer to see also a single plot with stacked bars representing the means per biotype.

A: A single plot of stacked bars has been added as a supplementary figure as suggested. We, however, preferred to keep figures 2 and 3 because we believe they provide important information for the manuscript. We had to split the samples and include the 'other' category because of the sheer number of biotopes/taxa and the difficulty in creating legible figures with so many different categories.

Figure4a-b: You could increase the area of plotting or reduce the font and circle size to better appreciate the cluster of biotypes on the lower left corner (Sed, Cor, HIG, HIX, Alg)

Figure 4a: I don't see the need in plotting different grey circle sizes to represent OTUs abundance, it's rather confusing.

A: In Fig 4a, the grey circles have been made less visible in order to increase the visibility of the sample symbols.

Figure 4b: Does circle size represent OTUs abundance?

A: The circles do indeed represent OTU abundance as mentioned in the legend.

Figure 5 and 6: I am hesitant about how these figures contribute to the line of the paper. If you are looking for indicator species, you shouldn't be restricting the search to the most abundant OTUs. Why didn't you represent e.g. SIMPER results instead? Or if you want to highlight key species, why not showing the abundance of Poribacteria across samples as an example of indicator species dynamics. I would rather send figures 5 and 6 to the supplementary data along with cutting down the text describing them.

A: Figures 5 and 6 have been included as supplementary figures in the revised version and a new

figure added based on the Simper results as suggested. The abundance of Poribacteria is also included in a supplementary figure.

Fig 7b-c need a better explanation. I would have expected to see more of this type of information. The idea of relating the number of OTUs and the biotypes in which OTUs were present is good and help to give some context of what proportion of OTUs are host-specific or shared.

A: The description in the legend for Fig 7b-c has been improved.

Other comments

I would have liked to see how many sequences per sample before and after quality check, as well as how many OTUs per sample before and after contamination removal.

A: We have provided more details on the quality control steps.

Figure 5: Please state that x-axis is showing OTU ID

A: The x-axis shows the biotope and the y-axis the OTU id number. This has been mentioned in the legend.

L88-91: Does “Sequences” refer to bacteria taxa or genes? Please specify.

A: The word “sequences” has been replaced by OTUs in order to avoid confusion.

L157: Replace “split” with “clustered”

A: This has been replaced as suggested.

L500: Add primer reference (Klindworth et al. 2013)

A: The Klindworth reference has been added as suggested.

L522, L525-26: I would combine the two phrases “OTUs were selected using USEARCH10 (Edgar 2013)” and “OTU clustering (97% sequence similarity threshold) was performed using the -cluster_otus command” into something similar to: “OTU clustering (97% sequence similarity threshold) was performed using the -cluster_otus command of USEARCH10 (Edgar 2013)”

A: The sentences have been changed as suggested.

L527: Can you specify what is your blank control? E.g., DNA extraction, PCR or sequencing blank

A: The blank control was a tube from the extraction kit where no tissue was added. No DNA was, furthermore, detected during the DNA control. This has been explained in the revised version.

L866: What do you mean by groups? biotypes? Please clarify.

A: The groups refers to biotopes and has been changed in the text.

Reviewer #2 (Remarks to the Author):

This paper combines the sequencing of 22 biotopes cohabiting coral reef systems in order to analyse

the level of overlap of microbial species (as OTUs) among them. The effort put in this study is valuable for the scientific community, and it is a novel result. Showing that microbial communities are largely shared across biotopes and that microbial communities of sponges are diverse but not the most diverse, as many times suggested, are important outcomes. These concepts can only really be explored with such large dataset as the one contained within this study.

The manuscript is well written and well organised, the methods are clearly explained and the conclusions are straight forward. However, the samples are not well described. Most come from two previous studies in Penghu island, but it is not easy to know which ones belong to either previous work. Moreover, samples are missing important metadata information, for instance, specific location of the sample within the archipelago, year of the sampling, depth of the samples taken, etc, which could be important factors affecting their results.

A: Most samples do not come from two previous studies, but are included for the first time in the present study. Only a small number of samples come from a previous study (Coelho et al. 2018). Metadata information has been added to Supplementary Table 1 as suggested.

I also think the analyses are superficial and general, and much more information and patterns could have been extracted from the data, mostly if the metadata is included, or by analysing more closely some groups. For instance, when a biotope shows different compositions (fig.4), there are no further analyses conducted that could explain those differences related to other factors (specific species, location, etc). Only in the case of LMA sponge, the differences were related to specific species, showing a group of species harbouring bacterial communities similar to seawater and other species including communities similar to host-related biotopes. That analysis could be interesting for nudibranchs (or others) as specified below. Also knowing that microbial communities can be very variable in temporal and spatial scales, these factors should be considered in any analysis.

A: As suggested, additional information has been provided about other biotopes including sea cucumbers, nudibranchs, algae and flatworms in order to explain in more detail the compositional differences observed.

For these reasons I have recommended a rejection of the manuscript, but I encourage a resubmission after some extra work is done.

To guide the authors in a resubmission, I have these comments:

Line 116. results and discussion?

A: Results has been changed to results and discussion.

Line 159. Change to "also included samples that were"

A: This has been changed as suggested.

Line 161. Move 'other' at the end of sentence. And in similar subsequent sentences.

A: This has been changed as suggested.

Line 173. What does it mean "most abundant" in here and in Fig 5 and 6? What is the threshold for the OTUs shown?

A: the phrase most abundant has been removed and Figures 5 and 6 moved to supplementary

information.

Lines 192 to 194. I believe this information still relates with figure 5. If so, Chloroflexi lineages are not "absent" in seawater, algae, or LMA sponges. As you say in line 174, chloroflexi seems present in all those but in low abundances. I can see a small dot in those groups. But I can not see is the ascidians biotope in figure 5.

A: The sentence in question did not refer to the present study, but to the study of Schmitt et al. (2011). It has been removed in order to avoid confusion.

Line 344. Maybe change "However, the similarity" to "This similarity"?

A: "However, the similarity" has been changed to "The similarity".

Lines 351 - 356. You suggest that the nudibranch gut bacteria is related to the preferred food, however, I am not sure that there was information about the specific situation of the nudibranch at the time of collection (i.e. sampled from the surface of a certain sponge species). I agree it would be interesting to see a study about the effects of diet in the nudibranch gut. But for the moment, I wonder if the authors have ruled out the possibility of the different gut communities being related to the nudibranch taxonomy or the sampling location (i.e. Phuket, Penghu, Pattaya, Koh Tao) for instance.

A: In the revised manuscript, we provide more information on the species of nudibranchs that included gut samples where the prokaryote communities closely resembled those of sponges.

Line 353. I think it refers to Fig 4.

A: This has been changed as suggested.

Line 362. Change to "including all OTUs with > 100 sequences"

A: This has been changed as suggested.

Line 369. Change to Fig 7d-h

A: This has been changed as suggested.

Line 369-383. This section needs rephrasing. Describe each diagram once, with all the information you want to say together.

A: The section has been rewritten as suggested.

Line 393. Change 'mean abundance' to 'total abundance' as it refers to the sum of all sequences. Change that in the supplemental table 6 where it says "Abundance" and in the table description says "Sum".

A: This has been changed as suggested.

Line 395. Clarify what 'this' refers to in "this also held".

A: This sentence has been removed.

Line 414 - 425. The host-associated endemism (or distribution) of microbial communities refers to bacteria/archaea found in significant numbers in a specific host. Of course, deeper sequencing will show more rare microorganisms being shared with other biotopes, but that does not contradict the apparent host-associated endemism the author refers to. For two reasons:

a) The well-studied host-associated community (of sponges for instance) is a result of evolution and selection in each host species to allow them to survive and grow to significant numbers, not just as 'rare' or transient bacteria.

b) Deeper sequencing will increase the arbitrary threshold of 100 sequences chosen in this work, but what is more important is the percentage they represent to the total sequences. That percentage would not be very affected with deeper sequencing. So, I think that sentence could be removed, or explained with more convincing arguments (i.e. how deeper sequencing of rare microorganism will contradict the host-associated specific communities).

A: The sentence about deeper sequencing has been removed as suggested.

Line 417. I am assuming you mean 'control' as control for contamination in line 417. The idea of sediments as control is valid, but it seems contradictory to use what seems to be a "seed bank" as control for contamination. If you want to discuss this part better, a more specific analysis including the sediment samples and all the other biotopes showing sediment sequences (not only in groups of 3), and including abundances, would be interesting.

A: This line has been removed and a more detailed analysis of the influence of sediment has been added.

Line 528. We have observed bleeding between samples from Illumina sequencing (reads being incorrectly assigned to samples) and thus are very critical of the QIIME contamination filtering approach. This method removes any OTU found in the blank/control from the dataset. We noticed that our most abundant reads have appeared in our blank controls (with very low counts, but > 0), most likely due to Illumina software/hardware issues, and using the QIIME filtering approach would remove true OTU from your dataset. Can you please comment on the number of reads and number of OTUs removed in this study, and clarify whether a significant amount of the data was removed?

A: We have provided more details on how contaminants were removed and modified our removal based on your observations.

Line 539-540. The use of "presence and abundance" is ambiguous, because in this field of work there are two approaches to analysing the data: presence/absence or abundances (counts). So it is unclear to the reader. Change to "OTU abundances or OTU counts".

A: The sentence has been changed as suggested (to OTU counts).

Line 866. Rephrase b). I did not understand that sentence.

A: The sentence has been rephrased as suggested.

Line 869 -874. Maybe change to: "d-h) Venn diagrams of the number of OTUs shared in groups of three biotopes. Abbr.: algae (Alg), holothurian gut (HiG), HMA sponges (HMA), sediment (Sed), stony corals (Cor), LMA sponges (LMA)".

A: The sentence has been changed as suggested.

Fig. 1 is not space efficient. The zoomed regions need to be positioned better to make use of the figure area. Also, I would change the dots in the general map by a square that represents the zoomed region.

Fig 1 has been removed and GPS coordinates included in Supplementary Table 1.

Fig 4 to 7. Why are only 13 biotopes instead of all 22 shown in these figures? This is not adequately explained in the text or caption.

A: In the revised version we assessed 21 biotopes in total, but grouped 10 biotopes together in 'others', thus creating 12 primary biotopes. This was done to improve the legibility of the figures. With the full 21 biotopes, the figures were too cramped and it was difficult to distinguish the various colours.

Fig 2 and fig 3.

- I believe this graph has been divided in two (13 biotopes fig 2 and 10 biotopes in fig3) because the rest of the plots only show the first 13 biotopes. If that is the case, maybe you can move Figure 3 to supplementary material and only keep the main 13 biotopes in the main text, to be consistent with the rest.

A: We prefer to maintain the figure just to give some indication of the values recorded for members of the 'other' category.

- I would order the biotopes as you named them in lines 122-127. Putting the 'Oth' at the end of the figure. And keep that other in all other figures (4 to 7).

A: We have moved 'Oth' to the end of the lists in the figures as suggested.

- I don't think you need the letters 'a to x'. The names are clear, and you don't need to explain them in the figure legend.

A: We prefer to maintain the letters in the figure and refer to specific subfigures later in the manuscript when we address the amount of environmental OTUs in host-associated prokaryote communities.

Fig 8. The colour combinations need to be improved. It is hard to associate the darker green to the overlapping of green and purple. Using some common 2 colour combination could work; blue + red = purple, or green + yellow = orange.

This figure has been omitted in the revised version.

Table S1. Even if samples are from previous papers, it would be helpful if you could still add in the table some of the metadata, at least specific location, date of collection and depths, and any other recorded information.

A: Metadata has been added as suggested.

Table S2. I would show the different families within Proteobacteria rather than just the whole phylum.

A: The main classes within the Proteobacteria have been added as suggested.

Table S6. Maybe it would be more informative to show the abundances in each biotype, instead of just presence/absence and a column with 'total abundance'.

A: We prefer to keep Table S6 as is so it can be directly compared to Fig. 5. We can add an additional Table including abundances if deemed necessary.

Reviewers' Comments:

Reviewer #1:

Remarks to the Author:

I like the revised manuscript and I think putting the data into a larger framework, but with the emphasis on sponges, benefits the narrative and conclusions. In particular, I commend stating the caveats in the conclusions section. I have no further comments.

Reviewer #2:

Remarks to the Author:

The authors have made considerable efforts to improve this version of the manuscript. Some of the previous issues are well resolved, however, since there have been major changes, some new parts need further revision.

General comments.

- A major concern for me is the category of samples called "other". I was not a fan of it in the first version, and I keep thinking that it does not make sense to compare all those samples as a group with the other well defined categories (e.g. nudibranch, sponges, etc). They tend to show intermediate situations, which is expected considering the very diverse mix of samples they represent.

I would recommend removal of that category (and the species included in it) from the manuscript or, if the authors consider them to be very valuable, then I would recommend to keep the species separated for the analyses and include those results/plots in supplementary material.

Personally, I cannot take home any message or conclusion from the "other" biotope, and believe this would extend to the scientific community.

- In one of your answers, you specified that: "We have provided more details on how contaminants were removed and modified our removal based on your observations". You have added the details in line 418 "Potential contaminants were removed from the OTU table if they occurred at least two times in a blank control".

If, as you say, there was a modification of the removal step (I believe from removal of any sequence occurring in the negative control to only the ones that occurred at least twice), how can the results be the same as the previous submission?

"We recorded 27678 OTUs assigned to 69 phyla over 2360000 sequences (after rarefying to 10000 sequences per sample" (Line 82) reports the same numbers as the previous version.

Moreover, I cannot see the details of how many sequences and OTUs you had before and after quality check and contamination removal.

- While the new terms (HDH and LDH) can be helpful to delineate groups within the scope of this study, I think it is clear that there is not going to be a dichotomy in diversity (as you wonder in line 137), but a gradient from low to high. Similar to HMA and LMA, a study could split samples based on their relative diversities, but that separation can be very different in another study using different species with a different range of diversities. To compare studies under that idea, a clear threshold should be determined, which has not been set. Could you specify what is the diversity limit you used or you suggest to separate HDH and LDH groups?

- Why did you decide to focus on the phylum level for most of the analyses?. For Fig 2, Phylum level only shows very large differences, but you could probably have extracted much more detail comparing Class or Order levels. Although you would need a different type of plots (allowing more compacted information).

- The sentence in the abstract ("over 99% of all OTUs with >100 sequences were present in multiple biotopes") is very shocking but a bit simplistic. Other than the percentage of shared OTUs, you could add information about how many sequences those OTUs represent (I stress this later on in the specific comments).

- From your results I also find important to state in the abstract something like:
" the most abundant OTUs (or dominant OTUs), despite being a minority, were biotope-specific, while other abundant OTUs were shared among biotopes, albeit in lower abundances".

You could also add in the abstract a sentence about the effect of the environment, the percentage of shared OTUs and the correspondent abundance of those ones, for instance for sponges and environment.

- Furthermore, in the last sentence, the "sponge-specific community" term does not imply sponge-exclusive communities any more. In fact, to avoid confusion, many papers are calling that as "sponge-enriched OTUs", to make clear that they are more abundant in sponges but can also be present in water or other biotypes in low abundances. In my opinion, that still makes the sponge composition unique to its host.

- Finally, as a general comment from your response to my previous review, there was no clear or specific information on the changes made, which forced me to have to constantly compare the versions. It would be easier if you could specify the new numbering of the figures, or lines where parts were changed, or more specifically how did you improve it, etc. to aid in the review process.

Specific comments

Fig 2-3. If the legend specifies that the plot shows "Mean relative abundance", I guess the plot should also say so for the Y labels. And then, make more clear in the plot that Fig. 2u and 2v are showing counts. Also the respective units for 2w and 2x.

In fact, plots 2s to 2v could be in another figure, they are showing a completely different analysis and information than the rest, and it gets complicated to visually separate that if they are all in the same figure.

The legend for 2u and 2v says "counts of OTUs100". The term "counts" is quite confusing because it is easily link to sequences rather than to OTUs. So, I recommend to recheck the text and keep a constant nomenclature where counts refers to 'sequence counts', and OTU are simple referred as 'number of OTUs'.

Also, at least in my pdf the text of F and P values are not easy to see.

Fig 5. Since you have some taxonomic assignment of those OTUs, on the left of the plot, you could add for instance the name of the Class, instead of only the OTU number. So the reader does not need to go to supplementary material to find that information.

Line 83 - 85: The number of OTUs for the specific samples has changed from the previous version, but the Table S1 has not changed. So I believe it was just a mistake in the text in the previous version, and that the correct number for sediments is 3503 and not the original 3491?

Line 138-139. I do not understand this sentence. What does higher taxa refer to?

Line 170-185. Did you check if the extra information you have (i.e. specific collection site, or depth) relates to any of the patterns you see, other than the species? When I asked for the metadata, which you have included in supplementary tables, I was hoping it could explain some of your patterns, but it seems you have not added that to your analyses, at least I do not see it in the results. I do not know if you tried.

Line 196. Remove the semicolon after the reference.

Line 230. What do you mean by " greater similarity of HMA ", similarity among all of them or among same species?

Line 232. "subset" of what? Could it just be "sets"?

Line 234. Add "the" in : Compare this to "the" prokaryote ...

Line 276. Fix "Fig. 56"

Line 262. Better give the abundance as percentage of the sequences. Same in line 266.

Line 295-296. Could you specify (as done in section "is everything everywhere") how many sequences those shared OTUs (i.e 93% of sponges with environment) represent?? Low? Maybe that can also be added to the plots Fig 2s to Fig 2v. Together with "counts of OTUs" (even better number of OTUs) and the "percentages of OTUs" you could add the "abundances of those OTUs", if it gets separated from Fig 2 as I suggested previously.

Line 330. Add a coma here: "sediment samples, and a subset"

Lines 376-379. Put sponges first since they have a larger dataset and it is the focus of your work, and then the rest by order of abundances.

Line 403. MRDNA should be spelled as MrDNA as in Line 419? In fact, I do not think you need to repeat that the control was sequenced at MrDNA in line 419, it seems obvious. And change "in a blank control" by "in the negative control", for consistency with line 403.

Lines 415. I don't usually like modifying the order of steps while describing the Methods. Reorder the sentence, so you avoid saying "prior to this".

Line 421. There are also some recent papers commenting about that, but I was expecting that the authors could make those observations by themselves, and discuss about it. Did you notice if the sequences present in the negative control were in general the most abundant ones in your other samples?

Some info: <https://www.biorxiv.org/content/early/2017/04/09/125724%20> and <https://www.illumina.com/content/dam/illumina-marketing/documents/products/whitepapers/index-hopping-white-paper-770-2017-004.pdf> for "index switching"

<https://www.ncbi.nlm.nih.gov/pubmed/25860802> and
<https://www.biorxiv.org/content/biorxiv/early/2018/05/28/332346.full.pdf> for "cross-talk"/"sample
bleeding"

Reviewers' comments:

Reviewer #1 (Remarks to the Author):

I like the revised manuscript and I think putting the data into a larger framework, but with the emphasis on sponges, benefits the narrative and conclusions. In particular, I commend stating the caveats in the conclusions section. I have no further comments.

Reviewer #2 (Remarks to the Author):

The authors have made considerable efforts to improve this version of the manuscript. Some of the previous issues are well resolved, however, since there have been major changes, some new parts need further revision.

General comments.

- A major concern for me is the category of samples called "other". I was not a fan of it in the first version, and I keep thinking that it does not make sense to compare all those samples as a group with the other well defined categories (e.g. nudibranch, sponges, etc). They tend to show intermediate situations, which is expected considering the very diverse mix of samples they represent.

I would recommend removal of that category (and the species included in it) from the manuscript or, if the authors consider them to be very valuable, then I would recommend

to keep the species separated for the analyses and include those results/plots in supplementary material.

Personally, I cannot take home any message or conclusion from the "other" biotope, and believe this would extend to the scientific community.

A: Although we find it regrettable, the 'other' category has been removed as suggested and the soft coral, chiton and sea urchin biotopes added. We understand the reviewer's reservations with the other category although it is not uncommon to use such a category when dealing with such a large number of taxa. It's unfortunate that not all the data could be used.

- In one of your answers, you specified that: "We have provided more details on how contaminants were removed and modified our removal based on your observations". You have added the details in line 418 "Potential contaminants were removed from the OTU table if they occurred at least two times in a blank control".

If, as you say, there was a modification of the removal step (I believe from removal of any sequence occurring in the negative control to only the ones that occurred at least twice), how can the results be the same as the previous submission?

"We recorded 27678 OTUs assigned to 69 phyla over 2360000 sequences (after rarefying to 10000 sequences per sample" (Line 82) reports the same numbers as the previous version.

A: This was an oversight. The correct number of OTUs and sequences has been reported in the revised version and in supplementary Table 1 for each sample separately.

Moreover, I cannot see the details of how many sequences and OTUs you had before and after quality check and contamination removal.

A: This information was not provided in the results section, but in the Material and Methods section including the number of sequences and OTUs removed as contaminants.

- While the new terms (HDH and LDH) can be helpful to delineate groups within the scope of this study, I think it is clear that there is not going to be a dichotomy in diversity (as you wonder in line 137), but a gradient from low to high. Similar to HMA and LMA, a study could split samples based on their relative diversities, but that separation can be very different in another study using different species with a different range of diversities. To compare studies under that idea, a clear threshold should be

determined, which has not been set. Could you specify what is the diversity limit you used or you suggest to separate HDH and LDH groups?

A: In Supplementary Figs 3 and 4, we explore this dichotomy. In the revised manuscript we make the discussion of the new terms less prominent. We also mention the continuous nature of Fig. 5. However, there were very large differences in OTU richness among species and have added new text (the first paragraph of 'HMA sponges are characterised by low richness but high evenness') with richness estimates of high, low and intermediate-richness species to demonstrate these differences.

- Why did you decide to focus on the phylum level for most of the analyses?. For Fig 2, Phylum level only shows very large differences, but you could probably have extracted much more detail comparing Class or Order levels. Although you would need a different type of plots (allowing more compacted information).

A: In the revised version, we included subfigures (in Fig. 2) showing class level analyses for Proteobacteria. In the first two versions we had decided to leave this out due to the number of graphs and limited amount of text available.

- The sentence in the abstract ("over 99% of all OTUs with >100 sequences were present in multiple biotopes") is very shocking but a bit simplistic. Other than the percentage of shared OTUs, you could add information about how many sequences those OTUs represent (I stress this later on in the specific comments).

A: We have added subfigures to a new figure showing the amount of sequences represented by the OTUs.

- From your results I also find important to state in the abstract something like: "the most abundant OTUs (or dominant OTUs), despite being a minority, were biotope-specific, while other abundant OTUs were shared among biotopes, albeit in lower abundances".

A: This would not be true. There were very few biotope-specific OTUs (only 21 of the 1731 OTUs₁₀₀) and these were not abundant. Some of the most abundant OTUs were, however, shared in only a few biotopes and were particularly abundant in only a single biotope as noted in the text.

You could also add in the abstract a sentence about the effect of the environment, the percentage of shared OTUs and the correspondent abundance of those ones, for instance for sponges and environment.

A: We have added this to the final sentence of the abstract. Unfortunately, the abstract length is only 150 words, which limits what we would like to say.

- Furthermore, in the last sentence, the “sponge-specific community” term does not imply sponge-exclusive communities any more. In fact, to avoid confusion, many papers are calling that as “sponge-enriched OTUs”, to make clear that they are more abundant in sponges but can also be present in water or other biotypes in low abundances. In my opinion, that still makes the sponge composition unique to its host.

A: The last sentence referring to the sponge-specific community was removed to make space for the sentence about shared OTUs between sponge hosts and the environment.

- Finally, as a general comment from your response to my previous review, there was no clear or specific information on the changes made, which forced me to have to constantly compare the versions. It would be easier if you could specify the new numbering of the figures, or lines where parts were changed, or more specifically how did you improve it, etc. to aid in the review process.

A: We apologize if changes were not clearly indicated in the previous version. In the present rebuttal, we have tried to be clearer on what has been altered. We have also uploaded a version with track changes so it can easily be seen what has been altered.

Specific comments

Fig 2-3. If the legend specifies that the plot shows “Mean relative abundance”, I guess the plot should also say so for the Y labels. And then, make more clear in the plot that Fig. 2u and 2v are showing counts. Also the respective units for 2w and 2x.

In fact, plots 2s to 2v could be in another figure, they are showing a completely different analysis and information than the rest, and it gets complicated to visually separate that if they are all in the same figure.

The legend for 2u and 2v says "counts of OTUs₁₀₀". The term “counts” is quite confusing because it is easily link to sequences rather than to OTUs. So, I recommend to recheck the text and keep a constant nomenclature where counts refers to 'sequence counts', and OTU are simple referred as 'number of OTUs'.

A: This has been modified as suggested and a new figure made for the OTU₁₀₀ data.

Also, at least in my pdf the text of F and P values are not easy to see.

A: This is legible in our pdf. We're not sure what the problem is. Perhaps the editor can verify.

Fig 5. Since you have some taxonomic assignment of those OTUs, on the left of the plot, you could add for instance the name of the Class, instead of only the OTU number. So the reader does not need to go to supplementary material to find that information.

A: Y-axis labels of the various Proteobacterial classes have been colour coded and this information has been added to the figure legend.

Line 83 - 85: The number of OTUs for the specific samples has changed from the previous version, but the Table S1 has not changed. So I believe it was just a mistake in the text in the previous version, and that the correct number for sediments is 3503 and not the original 3491?

A: We redid the analysis and everything has been checked and the correct number of OTUs has now been given in the manuscript and supplementary Table.

Line 138-139. I do not understand this sentence. What does higher taxa refer to?

A: Higher taxa refers to phyla, class level assessments etc.. This has been removed to avoid confusion.

Line 170-185. Did you check if the extra information you have (i.e. specific collection site, or depth) relates to any of the patterns you see, other than the species? When I asked for the metadata, which you have included in supplementary tables, I was hoping it could explain some of your patterns, but it seems you have not added that to your analyses, at least I do not see it in the results. I do not know if you tried.

A: We did some analyses, but the greatest part of variation was among biotopes. Residual information was largely due to species identity and HMA/LMA affiliation in the case of sponges. Geographical and environmental variation explained very little residual information compared the previous components as can indeed be seen in the PCO where samples of taxa from various sites, e.g., the sponge species *Xestospongia testudinaria*, grouped together according to species as opposed to sampling site. A more detailed study sampling the same species across the same set of sites and environments,

however, is required to accurately tease apart the relative contribution of biotope, species identity, geography and environment. This also goes beyond the scope of the present study where the goal was to compare different hosts/biotopes.

Line 196. Remove the semicolon after the reference.

A: This has been removed

Line 230. What do you mean by " greater similarity of HMA ", similarity among all of them or among same species?

A: This has been replaced by "the greater compositional similarity of the prokaryote communities of HMA as opposed to LMA sponges".

Line 232. "subset" of what? Could it just be "sets"?

A: Subsets has been replaced by sets as suggested.

Line 234. Add "the" in : Compare this to "the" prokaryote ...

A: 'the' has been added as suggested.

Line 276. Fix "Fig. 56"

A: This has been fixed.

Line 262. Better give the abundance as percentage of the sequences. Same in line 266.

A: The percentages have been added as suggested.

Line 295-296. Could you specify (as done in section "is everything everywhere") how many sequences those shared OTUs (i.e 93% of sponges with environment) represent?? Low? Maybe that can also be added to the plots Fig 2s to Fig 2v. Together with "counts of OTUs" (even better number of OTUs) and the "percentages of OTUs" you could add the "abundances of those OTUs", if it gets separated from Fig 2 as I suggested previously.

A: The percentages of sequences have been added as suggested to the text and in the new figure.

Line 330. Add a coma here: "sediment samples, and a subset"

A: A comma has been added as suggested.

Lines 376-379. Put sponges first since they have a larger dataset and it is the focus of your work, and then the rest by order of abundances.

A: The sponges have been put first as suggested.

Line 403. MRDNA should be spelled as MrDNA as in Line 419? In fact, I do not think you need to repeat that the control was sequenced at MrDNA in line 419, it seems obvious. And change "in a blank control" by "in the negative control", for consistency with line 403.

A: MRDNA has been removed and 'negative control' has been replaced by 'blank control'.

Lines 415. I don't usually like modifying the order of steps while describing the Methods. Reorder the sentence, so you avoid saying "prior to this".

A: The sentences have been reordered as suggested and 'prior to' removed.

Line 421. There are also some recent papers commenting about that, but I was expecting that the authors could make those observations by themselves, and discuss about it. Did you notice if the sequences present in the negative control were in general the most abundant ones in your other samples?

A: We have added two of the suggested articles to our manuscript and would like to thank the reviewer for suggesting them. We don't think that the present manuscript is a good place to discuss bleeding, but plan to address this in another manuscript where there are less word limitations.

Some info: <https://www.biorxiv.org/content/early/2017/04/09/125724%20> and <https://www.illumina.com/content/dam/illumina-marketing/documents/products/whitepapers/index-hopping-white-paper-770-2017-004.pdf> for "index switching"

<https://www.ncbi.nlm.nih.gov/pubmed/25860802> and
<https://www.biorxiv.org/content/biorxiv/early/2018/05/28/332346.full.pdf> for "cross-talk"/"sample bleeding"

Reviewers' Comments:

Reviewer #2:

Remarks to the Author:

I am satisfied with the changes made by the authors. I think the manuscript is now very clear, and the changes have clarified my confusion with the importance of the shared OTUs (by adding more OTU abundance information).

Even though it is hard to drop samples that involved money and effort, sometimes is necessary if they cannot provide conclusive results and are only going to add noise. The 'other' sample group was not very useful for comparison with future studies, however now the manuscript includes more biotopes that extend the description of the coral reef.

A last comment, could you please add info of the specific plot (a, b, etc) for Figures 2 and 3 in the main text, as you did for Figs. 5 and 6.

Reviewers' Comments:

Reviewer #2:

Remarks to the Author:

I am satisfied with the changes made by the authors. I think the manuscript is now very clear, and the changes have clarified my confusion with the importance of the shared OTUs (by adding more OTU abundance information).

Even though it is hard to drop samples that involved money and effort, sometimes is necessary if they cannot provide conclusive results and are only going to add noise. The 'other' sample group was not very useful for comparison with future studies, however now the manuscript includes more biotopes that extend the description of the coral reef.

A last comment, could you please add info of the specific plot (a, b, etc) for Figures 2 and 3 in the main text, as you did for Figs. 5 and 6.

A: Info about the specific subplots has been added the the main text as suggested.